# Multi-Robot Motion Planning with Diffusion Models

**Yorai Shaoul**[*, 1]**, Itamar Mishani**[*, 1]**, Shivam Vats**[*, 2]**, Jiaoyang Li**[1] **& Maxim Likhachev**[1]
[1]Carnegie Mellon University    [2]Brown University    [*]Equal contribution
{`yshaoul,imishani,svats,jiaoyanl,maxim`}@cs.cmu.edu

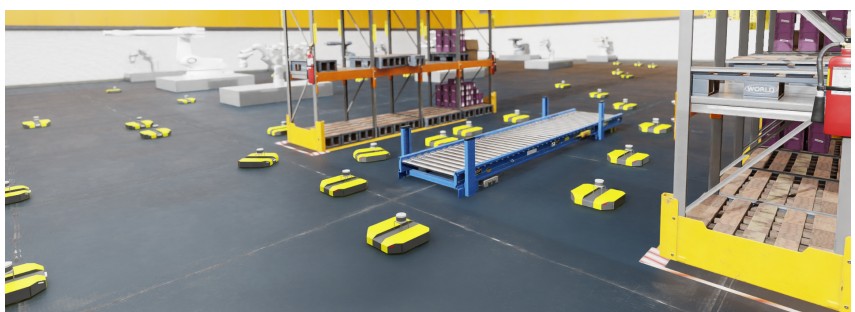

## Abstract

Diffusion models have recently been successfully applied to a wide range of robotics applications for learning complex multi-modal behaviors from data. However, prior works have mostly been confined to single-robot and small-scale environments due to the high sample complexity of learning multi-robot diffusion models. In this paper, we propose a method for generating collision-free multi-robot trajectories that conform to underlying data distributions while using only single-robot data. Our algorithm, Multi-robot Multi-model planning Diffusion (MMD), does so by combining learned diffusion models with classical search-based techniques—generating data-driven motions under collision constraints. Scaling further, we show how to compose multiple diffusion models to plan in large environments where a single diffusion model fails to generalize well. We demonstrate the effectiveness of our approach in planning for dozens of robots in a variety of simulated scenarios motivated by logistics environments.

## 1 Introduction

Multi-robot motion planning (MRMP) is a fundamental challenge in many real-world applications where teams of robots have to work in close proximity to each other to complete their tasks. In automated warehouses, for example, hundreds of mobile robots and robotic manipulators need to coordinate with each other to transport and exchange items while avoiding collisions. Learning motions from demonstrations can oftentimes allow robots to complete tasks they couldn't otherwise, like navigating a region in a pattern frequently followed by human workers; however, it is unclear how to best incorporate demonstrations in MRMP. In fact, MRMP at its simplest form, where robots are only concerned with finding short trajectories between start and goal configurations, is already known to be computationally intractable (Hopcroft & Wilfong, 1986)—significantly harder than single-agent motion planning due to the complexity of mutual interactions between robots. Attempting to simplify the problem, various approximate formulations have been proposed in the literature. For example, a popular approach is to formulate the problem as a multi-agent path finding problem (MAPF) (Stern et al., 2019) by discretizing space and time. While the latest MAPF planners (Li et al., 2021; Okumura, 2024) can compute near-optimal plans and scale to hundreds of agents, they make strong assumptions, such as constant velocities and rectilinear movements that limit their real-world application and reduce their ability to generate complex behaviors learned from demonstrations.

In single-agent motion planning, methods that learn to plan from data (Xiao et al., 2022) have been widely used to circumvent similar limitations resulting from inaccurate models (Vemula et al., 2021),

partial observability (Choudhury et al., 2018) and slow planning (Sohn et al., 2015; Qureshi et al., 2020). More recently, diffusion models (DM) have emerged as the generative model of choice for learning visuomotor manipulation policies from demonstrations (Chi et al., 2024), motion planning (Carvalho et al., 2023), and reinforcement learning (Janner et al., 2022). However, there has been relatively little work on extending these ideas to multi-robot motion planning. This is due to the twin challenges of generating high quality multi-agent data and the *curse of dimensionality*, i.e., significantly higher sample complexity of learning multi-robot models.

In this paper, we propose a data-efficient and scalable multi-robot diffusion planning algorithm, **M**ulti-robot **M**ulti-model planning **D**iffusion (MMD), that addresses both these challenges by combining constraint-based MAPF planners with diffusion models. Importantly, our approach calls for learning only *single-robot diffusion models*, which does away with the difficulty of obtaining multi-robot interaction data and breaks the curse of dimensionality. MMD generates collision-free trajectories by *constraining* single-robot diffusion models using our novel spatio-temporal guiding functions and choosing constraint placement via strategies inspired by MAPF algorithms. Our contributions in this paper are threefold: (1) We propose a novel data-efficient framework for multi-robot diffusion planning inspired by constraint-based search algorithms. (2) We provide a comparative analysis of the performance of five MMD algorithms, each based on a different MAPF algorithm, shedding light on their applicability to coordinating robots leveraging diffusion models for planning. (3) We show that we can scale our approach to arbitrarily large and diverse maps by learning and composing multiple diffusion models for each robot. Our experimental results from varied motion planning problems in simulated scenarios motivated by logistics environments suggest that our approach scales favorably with both the number of agents and the size of the environment when compared to alternatives. Video demonstrations and code are available at `https://multi-robot-diffusion.github.io/`.

## 2 PRELIMINARY

In this section we define the MRMP problem and provide relevant background on constraint-based MAPF algorithms and on planning with diffusion models. Sec. 3 elaborates on how we combine these concepts to coordinate numerous robots that plan with diffusion models.

### 2.1 MULTI-ROBOT MOTION PLANNING (MRMP)

Given $n$ robots $\mathcal{R}_i$, MRMP seeks a set of collision-free trajectories, one for each robot, that optimize a given objective function. Let $\mathcal{S}^i$ be the state space of a single robot and a state be $\mathbf{s}^i := [\mathbf{q}^i, \dot{\mathbf{q}}^i]^\mathsf{T} \in \mathcal{S}^i$ where $\mathbf{q}^i$ and $\dot{\mathbf{q}}^i$ are the configuration and velocity of the robot. Each robot has an assigned start state $\mathbf{s}^i_{\text{start}} \in \mathcal{S}^i$ and binary termination (goal) condition $\mathcal{T}^i : \mathcal{S}^i \rightarrow \{0, 1\}$. An MRMP solution is a multi-robot trajectory $\boldsymbol{\tau} = \{\boldsymbol{\tau}^1, \cdots \boldsymbol{\tau}^n\}$, where $\boldsymbol{\tau}^i : [0, T^i] \rightarrow \mathcal{S}^i$ represents the trajectory of robot $\mathcal{R}_i$ over the time interval $[0, T^i]$, with $T^i$ being the terminal time. In practice, we uniformly discretize the time horizon into $H$ time steps and optimize over a sequence of states $\boldsymbol{\tau}^i = \{\mathbf{s}^i_1, \mathbf{s}^i_2, ..., \mathbf{s}^i_H\}$. Subscripts, e.g., $\boldsymbol{\tau}^i_t$, indicate indexing into a trajectory. Each trajectory $\boldsymbol{\tau}^i$ must avoid collisions between robots and with obstacles in the environment. In MRMP, robots share a workspace $\mathcal{W}$ (i.e., $\mathcal{W} \subseteq \mathbb{R}^3$ for general robots and $\mathcal{W} \subseteq \mathbb{R}^2$ for robots on the plane) and occupy some volume or area within $\mathcal{W}$, which we denote as $\mathcal{R}_i(\mathbf{q}^i) \subseteq \mathcal{W}$ for robot $\mathcal{R}_i$ in configuration $\mathbf{q}^i$.

The usual MRMP objective is to minimize the sum of the single-robot costs (e.g., the cumulative motion) across all robots. General cost functions can be defined on the trajectories, and the objective then becomes $\mathcal{J}(\boldsymbol{\tau}) = \frac{1}{n} \sum_{i=1}^{n} \text{cost}(\boldsymbol{\tau}^i)$. When learning from data, we are interested in *data adherence*, i.e., the trajectories should match the underlying trajectory distribution. We define $\text{cost}_{\text{data}}(\boldsymbol{\tau}) = \frac{1}{n} \sum_{i=1}^{n} \text{cost}_{\text{data}}(\boldsymbol{\tau}^i)$ to quantify how well, on average, trajectories in $\boldsymbol{\tau}$ follow their underlying data distribution. This metric is task-specific; we provide some examples in Sec. 4.

### 2.2 MULTI-AGENT PATH FINDING (MAPF)

The MAPF problem, a simpler form of MRMP, seeks the shortest collision-free *paths* $\Pi = \{\boldsymbol{\pi}^1, \boldsymbol{\pi}^2, \cdots \boldsymbol{\pi}^n\}$ for $n$ agents on a graph. This graph approximates their configuration space, with vertices corresponding to configurations and edges to transitions. Each *path* $\boldsymbol{\pi}^i = \{\mathbf{q}^i_1, \cdots \mathbf{q}^i_H\}$ is a trajectory without velocity that need not be dynamically feasible. In MAPF studies, constraint-based

algorithms have become popular due to their simplicity and scalability. These algorithms are effective, in part, because they avoid the complexity of the multi-agent configuration space by delegating planning to single-agent planners and avoid collision via constraints. For instance, if a configuration $\mathbf{q}^i$ for $\mathcal{R}_i$ leads to a collision at time (or interval) $t$, this can be prevented by applying the constraint set $C = \{\langle \mathcal{R}_i, \mathbf{q}^i, t \rangle\}$ to the path $\pi^i$, thereby preventing the configuration from being used at that time. Several MAPF algorithms, including Prioritized Planning (PP) (Erdmann & Lozano-Perez, 1987) and Conflict-Based Search (CBS) (Sharon et al., 2015), use this mechanism to force single-agent planning queries to avoid states that would lead to collisions. We detail these methods in Sec. 3 and explain how, despite traditionally being used for MAPF, their principles can be applied to coordinating robots in continuous space that generate data-driven trajectories via diffusion models.

## 2.3 PLANNING WITH DIFFUSION MODELS

Motion planning diffusion models are generative models that learn a denoising process to recover a dynamically-feasible trajectory from noise (Carvalho et al., 2023; Janner et al., 2022). Given a dataset of multi-modal trajectories, diffusion models aim to generate new trajectories that follow the underlying distribution of the data. Additionally, these trajectories may be conditioned on a task objective $\mathcal{O}$, for example, goal condition and collision avoidance. Specifically, given a task objective $\mathcal{O}$, motion planning diffusion models aim to sample from the posterior distribution of trajectories:

$$\arg\max_{\boldsymbol{\tau}^i} \log p(\boldsymbol{\tau}^i | \mathcal{O}) = \arg\min_{\boldsymbol{\tau}^i} \left( \mathcal{J}(\boldsymbol{\tau}^i) - \log p(\boldsymbol{\tau}^i) \right) \tag{1}$$

The first term of the objective, $\mathcal{J}(\boldsymbol{\tau}^i)$, can be interpreted as a standard motion planning objective (Carvalho et al., 2023), in which we try to minimize a cost function (or, equivalently, maximize a reward function). The second term, $\log p(\boldsymbol{\tau}^i)$, is the prior corresponding to the data adherence discussed in Sec. 2.1.

Diffusion models are a type of score-based model (Song et al., 2021), where the focus is on learning the score function (the gradient of the data distribution's log-probability) rather than learning the probability distribution directly. The score function is learned using *denoising score matching*, a technique for learning to estimate the score by gradually denoising noisy samples. The diffusion inference process consists of a $K$-step denoising process that takes a noisy trajectory ${}^K\boldsymbol{\tau}^i$ and recovers a feasible trajectory ${}^0\boldsymbol{\tau}^i$, which also follows the data distribution. We use the notation ${}^0\boldsymbol{\tau}^i, {}^1\boldsymbol{\tau}^i, \cdots, {}^K\boldsymbol{\tau}^i$ to denote the evolution of the trajectory in the diffusion process. To generate a trajectory ${}^0\boldsymbol{\tau}^i$ from a noise trajectory ${}^K\boldsymbol{\tau}^i \sim \mathcal{N}(\mathbf{0}, \mathbf{I})$, we use Langevin dynamics sampling (Ho et al., 2020), an iterative process that is a type of Markov Chain Monte Carlo method. At each denoising step $k \in \{K, \ldots, 1\}$, a trajectory-space mean $\mu_{k-1}^i$ is sampled from the network $\mu_\theta$:

$$\mu_{k-1}^i = \mu_\theta({}^k\boldsymbol{\tau}^i) \tag{2}$$

Now, with the variance prescribed by a deterministic schedule $\{\beta_k \mid k \in \{K, \cdots, 1\}\}$, the next trajectory in the denoising sequence is sampled from the following distribution:

$$
{}^{k-1}\boldsymbol{\tau}^i \sim \mathcal{N}\big(\mu_{k-1}^i + \underbrace{\eta\beta_{k-1}\nabla_{\boldsymbol{\tau}}\mathcal{J}(\mu_{k-1}^i)}_{\text{Guidance}}, \beta_{k-1}\big) \tag{3}
$$

The term $\nabla_{\boldsymbol{\tau}}\mathcal{J}(\mu_{k-1}^i)$ is the gradient of additional trajectory-space objectives (described in Eq. 1) imposed on the generation process. This term, also called *guidance*, can include multiple weighted cost components, each optimizing for a different objective. For instance, we can have $\mathcal{J} = \lambda_{\text{obj}}\mathcal{J}_{\text{obj}} + \lambda_{\text{smooth}}\mathcal{J}_{\text{smooth}}$ to penalize trajectories in collision with objects via $\mathcal{J}_{\text{obj}}$ and to encourage the trajectory to be dynamically feasible via $\mathcal{J}_{\text{smooth}}$. We denote the trajectory generation process queried with a start state $\mathbf{s}_{\text{start}}^i$, goal condition $\mathcal{T}^i$, and costraint set $C$ (Sec. 3.1) as $f_\theta^i(\mathbf{s}_{\text{start}}^i, \mathcal{T}^i, C)$.

## 3 METHOD

We present Multi-robot Multi-model planning Diffusion (MMD), an algorithm for flexibly scaling diffusion planning to multiple robots and long horizons using only single-robot data. MMD imposes constraints on diffusion models to generate collision-free trajectories, addressing three main questions: *how*, *when*, and *where* to impose them. First, we discuss integrating spatio-temporal constraints into the diffusion denoising process through guiding functions. Next, we introduce five MMD algorithms, each inspired by a MAPF algorithm regarding constraint placement and timing. Finally, we demonstrate how to sequence multiple models for long-horizon planning.

**Algorithm 1:** MMD-CBS sketch. Colored lines are only in MMD-PP, MMD-ECBS

**Input:** Starts, goal conditions, and single-robot diffusion models $\left\{\mathbf{s}_{\text{start}}^i, \mathcal{T}^i, f_\theta^i\right\}_{i=1}^n$
**Output:** Trajectories $\boldsymbol{\tau} = \left\{\boldsymbol{\tau}^i\right\}_{i=1}^n$

$N_{\text{root}} \leftarrow$ new CT node; $N_{\text{root}}.C^i \leftarrow \emptyset \; \forall i \in \{1, \cdots n\}$
**for** $i \in \{1, \cdots, n\}$ **do**
 $C_{\text{strong}}^i, C_{\text{weak}}^i \leftarrow \emptyset, \emptyset$    // Empty constraints set.
 $C_{\text{strong}}^i \leftarrow \{\langle \mathcal{R}_i, N_{\text{root}}.\boldsymbol{\tau} \rangle\}$    // Avoid other robots.
 $C_{\text{weak}}^i \leftarrow \{\langle \mathcal{R}_i, N_{\text{root}}.\boldsymbol{\tau} \rangle\}$    // Penalize collisions.
 $N_{\text{root}}.\boldsymbol{\tau}^i \leftarrow f_\theta^i(\mathbf{s}_{\text{start}}^i, \mathcal{T}^i, C_{\text{strong}}^i \cup C_{\text{weak}}^i)$
**end**
**return** $N_{\text{root}}.\boldsymbol{\tau}$
$\text{CT} \leftarrow \{N_{\text{root}}\}$    // Initialize CT.
**while** $\text{CT} \neq \emptyset$ **do**
 $N \leftarrow \underset{N' \in \text{CT}}{\arg\min} \, \texttt{numConflicts}(N'.\boldsymbol{\tau})$
 Remove $N$ from CT
 **if** $N.\boldsymbol{\tau}$ *conflict-free* **then**
  **return** $N.\boldsymbol{\tau}$    // Return if collision free.
 **end**
 $p, t, \mathcal{R}_i, \mathcal{R}_j \leftarrow \texttt{getOneConflict}(N.\boldsymbol{\tau})$
 **for** $k \in \{i, j\}$   // Split $N$; constrain conflicting robots. **do**
  $N' \leftarrow N.\text{copy}$
  $N'.C^k \leftarrow C^k \cup \{\langle \mathcal{R}_k, S_r(p), \mathbf{t} \rangle\}$
  $C_{\text{weak}}^k \leftarrow \{\mathcal{R}_k, \langle N'.\boldsymbol{\tau} \rangle\}$    // Penalize collisions.
  $N'.\boldsymbol{\tau}^k \leftarrow f_\theta^k(\mathbf{s}_{\text{start}}^k, \mathcal{T}^k, N'.C^k \cup C_{\text{weak}}^k)$
  $\text{CT} \leftarrow \text{CT} \cup \{N'\}$
 **end**
**end**

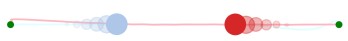

(a) Two robots aim to switch positions. Blindly generated single-robot trajectories collide.

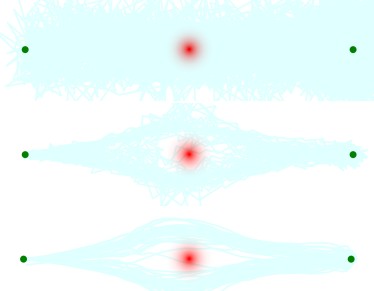

(b) The diffusion denoising process for the left robot in (a), under a temporally-activated constraint (in red), yields multi-modal trajectories.

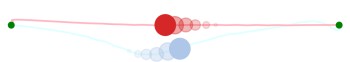

(c) Collision-free solution.

Figure 1: An illustration of how MMD-CBS generates collision-free trajectories with constrained diffusion models.

## 3.1 CONSTRAINTS IN DIFFUSION MODELS

An intuitive and effective constraint for multi-robot motion planning in robotics is the *sphere constraint* [1] (Li et al., 2019; Shaoul et al., 2024b). It is defined by a point $p \in \mathcal{W}$ and restricts robots from being closer to $p$ than a radius $r \in \mathbb{R}$ at a certain time range $\mathbf{t} := [t - \Delta t, t + \Delta t]$. The sphere constraint can be *imposed as a soft-constraint* on a diffusion model by incorporating it in its guiding function $\mathcal{J}(\cdot)$. This can be done by adding a cost term $\mathcal{J}_c$ that repels the robots from the sphere's center point $p$. Let ${}^k\boldsymbol{\tau}^i$ be the generated trajectory for $\mathcal{R}_i$ at step $k$ of the diffusion denoising process, and $\langle \mathcal{R}_i, S_r(p), \mathbf{t} \rangle$ be a sphere constraint centered at $p$ with radius $r$ over time interval $\mathbf{t}$. The guidance cost term for $\mathcal{R}_i$ can be defined as:

$$\mathcal{J}_c({}^k\boldsymbol{\tau}_t^i) := \sum_{t \in \mathbf{t}} \max\left(\epsilon \cdot r - d\left(\mathcal{R}_i({}^k\boldsymbol{\tau}_t^i), p\right), \, 0\right) \quad (4)$$

with $d\left(\mathcal{R}_i({}^k\boldsymbol{\tau}_t^i), p\right)$ as the distance from point $p$ to $\mathcal{R}_i$ at ${}^k\boldsymbol{\tau}_t^i$, and $\epsilon \geq 1$ a padding factor.

## 3.2 CONSTRAINING STRATEGIES

To determine *when* and *where* to apply constraints on diffusion models, MMD draws on MAPF strategies like CBS and PP. We propose five MMD variants, each inspired by a state-of-the-art search algorithm. Alg. 1 provides a summary of these methods and we elaborate upon them here [2].

---

[1] The sphere constraint generalizes the MAPF vertex constraint, as it constrains robots from visiting the point of collision itself instead of a single colliding configuration corresponding to a vertex in a graph. In MAPF, the point of collision and the graph vertex coincide.

[2] In MMD, we use the search or prioritization logic found in MAPF algorithms for placing "strong" constraints on diffusion models, while all other aspects of MMD are more loosely inspired by MAPF algorithms.

**MMD-PP.** Prioritized Planning sequentially plans *paths* for robots $\mathcal{R}_i \; \forall \; i \in \{1, \ldots n\}$. This ordering of robots is treated as a priority ordering in that, on each call, robot $\mathcal{R}_i$ must generate a path $\boldsymbol{\pi}^i$ that avoids all $\mathcal{R}_j$ that previously planned. Robot $\mathcal{R}_i$ does so by respecting the constraints $C := \{\langle \mathcal{R}_i, \mathbf{q}^i, t \rangle \mid \mathcal{R}_i(\mathbf{q}^i) \cap \mathcal{R}_j(\boldsymbol{\pi}_t^j) \neq \emptyset \; \forall t\}$. To translate this approach to *trajectory* generation with diffusion models, *MMD-PP* represents robot volumes using spheres, as is common in robotics, and uses the sphere representation of higher-priority robots as sphere soft-constraints for lower-priority robots. Specifically, let a high-priority robot $\mathcal{R}_j$ be modeled with $M_j$ spheres and $p_m^j$ and $r_m^j$ be the position and radius of the $m^{\text{th}}$ sphere at time $t$. Then, lower-priority robot $\mathcal{R}_i$ generates a trajectory under the constraint set $\{\langle \mathcal{R}_i, S_{r_m^j}(p_m^j), t \rangle \mid m \in \{1, \cdots M_j\}, j \prec i\}$, where $\prec$ indicates priority precedence. In Alg. 1, $\langle \mathcal{R}_i, \boldsymbol{\tau} \rangle$ means that all trajectories of $\mathcal{R}_{j \neq i}$ in $\boldsymbol{\tau}$ must be similarly avoided.

**MMD-CBS.** CBS is a popular MAPF solver that combines "low-level" planners for individual agents with a "high-level" constraint tree (CT) to resolve conflicts (i.e., collisions). The algorithm initiates by creating the root node $N_{\text{root}}$ in the CT, planning paths for each agent independently, and storing these paths in $N_{\text{root}}.\Pi$. CBS repeatedly extracts nodes $N$ from the CT and inspects $N.\Pi$ for conflicts. If no conflicts exist, the algorithm terminates, returning $N.\Pi$. Otherwise, CBS selects a conflict time $t$ where agents $\mathcal{R}_i$ and $\mathcal{R}_j$ collide at positions $\mathbf{q}^i = \boldsymbol{\pi}_t^i$ and $\mathbf{q}^j = \boldsymbol{\pi}_t^j$ in $N.\Pi$. CBS then splits node $N$ into two new CT nodes, $N_i$ and $N_j$, each inheriting the (initially empty) constraint set $N.C$ and paths $N.\Pi$ from $N$, and incorporating a new constraint for preventing the respective agent from occupying the conflict position at time $t$. For example, $N_i.C \leftarrow N.C \cup \{\langle \mathcal{R}_i, \mathbf{q}^i, t \rangle\}$ for $\mathcal{R}_i$. Paths for $\mathcal{R}_i$ and $\mathcal{R}_j$ are then replanned using low-level planners under the updated constraints in $N_i.C$ and $N_j.C$. The new CT nodes, with updated paths in $N_i.\Pi$ and $N_j.\Pi$, are added to the CT. *MMD-CBS* follows the general CBS structure. It keeps a CT of nodes $N$, each with trajectories $N.\boldsymbol{\tau}$, and uses motion planning diffusion models as low-level planners. The algorithm identifies a *collision point* $p$ for each conflict and resolves it by imposing sphere soft-constraints centered at $p$ on affected robots (see Fig. 1 for an illustration and Sec. A.4 for parameter values).

**MMD-ECBS.** Enhanced-CBS (ECBS) (Barer et al., 2014) informs CBS low-level planners of the paths of other robots in the same CT node and steers the search towards solutions that are more likely to be collision-free. To emulate this in diffusion-based trajectory generation, MMD-ECBS imposes two types of soft constraints: "weak" and "strong." For each robot $\mathcal{R}_j$ with a trajectory in the CT node $N$, a weak soft-constraint that forbids $\mathcal{R}_i$ from colliding with any other $\mathcal{R}_j$ with $\boldsymbol{\tau}^j \in N.\boldsymbol{\tau}$ is imposed. This is done in a similar way to MMD-PP but with a lower penalty value (Sec. A.4). The strong constraints are the same as those in MMD-CBS, resolving previously observed conflicts.

**Reusing Experience in CBS.** Recent studies indicate that leveraging previous single-robot solutions to guide replanning enhances the efficiency of CBS (Shaoul et al., 2024a). This is primarily because the motion planning problem between a CT node and its successors is nearly identical, with the only difference being a single constraint, making planning from scratch wasteful. This can be utilized in MMD replanning by initially adding noise to the stored trajectory for a limited number of steps (3 in our experiments; regular inference uses 25 steps) and then denoising with the new soft-constraints. This approach, in the context of single-robot planning, was first proposed in Janner et al. (2022) and further refined in Zhou et al. (2024). Adding this functionality to MMD-CBS and MMD-ECBS yields our two final MMD algorithms, **MMD-xCBS** and **MMD-xECBS**, respectively. Both reuse previous solutions to inform replanning and are otherwise unchanged.

### 3.3 SEQUENCING DIFFUSION MODELS FOR LONG HORIZON PLANNING

Diffusion models have shown notable success in learning trajectory distributions within specific contexts. However, they face challenges in modeling complex trajectory distributions and generalizing to diverse contexts (e.g., significantly different obstacle layouts). We propose utilizing an ensemble of local diffusion models for each robot to facilitate varying-context planning. Each local model is trained to capture a particular motion pattern, i.e., a trajectory distribution generated by a hidden cost function defined by a specific task dataset. For example, near a conveyor belt, we can define a motion pattern requiring robots to pass through either the top corridor right-to-left, or the bottom corridor left-to-right. By sequentially combining multiple local models, each corresponding to a local map segment, we enable long-horizon single-robot planning that is easier to learn, generalizes well to different contexts, and scales effectively to large maps.

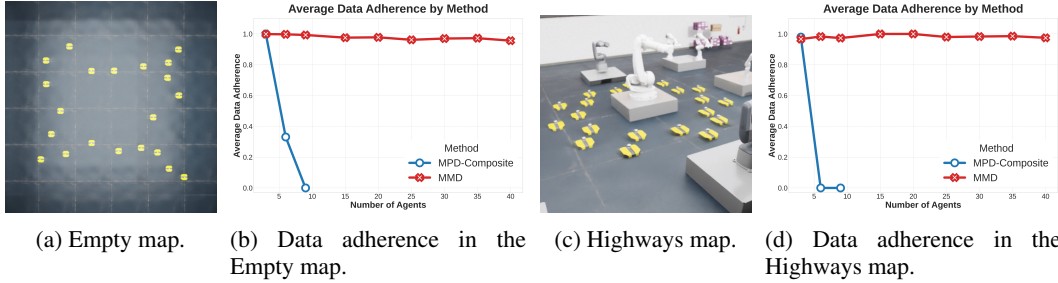

(a) Empty map.  (b) Data adherence in the Empty map.  (c) Highways map.  (d) Data adherence in the Highways map.

Figure 2: A comparison between MMD and "composite" diffusion models that generate trajectories for all agents at once. We observed consistent performance from MMD but a sharp decrease for the baseline, unable to produce valid solutions for 9 agents (denoted as zero adherence score). Since MMD uses the same single-agent model for all robots in these experiments, it is straightforward to keep increasing the number of agents without needing any retraining or new datasets.

During online planning, given a specific single-robot task $\mathcal{O}^i$ (which includes the start state and the termination condition), we can query the task-relevant local diffusion models to generate a full-horizon motion plan. Motivated by Mishra et al. (2023), this is done by sampling from the local models in parallel, while incorporating a *cross-conditioning* term that constrains the local trajectories to connect seamlessly. Let $\boldsymbol{\tau}^{i,l}$ be a local trajectory sampled from a local probability distribution $p^l$. The goal of the single-robot planner is to sample from the posterior distribution of trajectories:

$$\underset{\boldsymbol{\tau}^i}{\arg\max} \qquad \log p(\boldsymbol{\tau}^i|\mathcal{O}) = \underset{\boldsymbol{\tau}^{i,1},\ldots,\boldsymbol{\tau}^{i,L}}{\arg\max} \log p(\boldsymbol{\tau}^{i,1},\ldots,\boldsymbol{\tau}^{i,L}|\mathcal{O}) \tag{5a}$$

$$\text{subject to} \qquad \boldsymbol{\tau}_1^{i,l} = \boldsymbol{\tau}_{H_{l-1}}^{i,l-1}, \forall l = 1,\ldots,L \tag{5b}$$

Following Eq. 1, the new objective becomes:

$$\underset{\boldsymbol{\tau}^{i,1},\ldots,\boldsymbol{\tau}^{i,L}}{\arg\min} \left( \mathcal{J}(\boldsymbol{\tau}^{i,1},\ldots,\boldsymbol{\tau}^{i,L}) - \sum_{l=1}^{L} \log p(\boldsymbol{\tau}^{i,l}) \right) \tag{6}$$

In practice, MMD ensures proper sequencing of the $L$ local diffusion models by introducing constraints requiring the last state of the trajectory from model $l$ to be equal to the first state from model $l+1$ (see Eq. 5b) and treating generation of local trajectories as inpainting (Lugmayr et al., 2022). [3]

## 4 EXPERIMENTAL ANALYSIS

We tested MMD's efficacy in learning multi-robot trajectories on increasingly complex maps with varying numbers of holonomic ground robots in a simulated warehouse, modeling robots as 2D disks. Our goals were to (i) compare our approach with common methods for integrating data into multi-robot trajectory generation, (ii) identify the most effective constraining strategies with MMD, and (iii) evaluate MMD's scalability. Each experiment with $n$ robots begins by randomly picking start and goal states on a map for various algorithms to compute valid trajectories $\boldsymbol{\tau}$ (or MAPF paths $\Pi$) between. We evaluated the methods by *success rate*, the percentage of problems solved with no collision within a time limit, and *data adherence*, the average alignment of $\boldsymbol{\tau}^i \in \boldsymbol{\tau}$ with the dataset motion patterns. Data adherence uses a map-specific function $\text{cost}_{\text{data}}(\boldsymbol{\tau}^i)$, scoring 1 for perfect conformity and lower otherwise. Our evaluation maps, datasets, and adherence functions are summarized here with simple illustrations and more details are in the appendix.

---

[3]Another option is to add a guidance term penalizing discontinuity. This approach is more flexible as it can optimize for connection points that are learned as well (e.g., via classifier guidance on the connection itself).

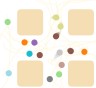

**Drop-Region** map (Fig. 3c) simulates package drop-off chutes. Motion demonstrations are trajectories between random states that include a pause at one of sixteen drop-off regions—next to any chute edge midpoint. Adherence is met for $\tau^i$, i.e., $\text{cost}_{\text{data}}(\tau^i) = 1$, if it includes such a pause. Otherwise, it is zero.

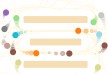

**Conveyor** map (Fig. 3b) features narrow passages with directional motion requirements. Demonstrations connect random states with trajectories that pass along the top corridor to the left, or through the bottom corridor to the right. Trajectory $\tau^i$ adheres to data if it similarly passes through either corridor before reaching its goal.

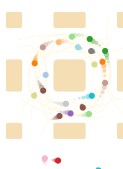

**Highways** map (Fig. 3a), requires counter-clockwise movement around a central obstacle—a pattern shown in its associated data. This map can be seen as a modular building block for larger multi-robot environments with its required motion pattern promoting easier coordination. Adherence is met for $\tau^i$ if its cumulative motion within the map is counter-clockwise.

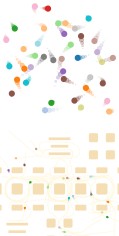

**Empty** map (Fig. 2a) is our simplest. Robots are scored highly if they move in straight lines and gradually worse the more they deviate. Demonstrations are of straight line motions.

**Larger Maps** (Fig. 5). In our larger scale experiments (Sec. 4.3) trajectories are computed within $2 \times 2$ and $3 \times 3$ maps composed of smaller, local maps. Required motions are dictated by spanned local maps, and the overall adherence is the average adherence per local map.

## 4.1 DECOUPLING SCALES MULTI-ROBOT DIFFUSION PLANNING

An appealing approach for learning multi-robot trajectory generation is by obtaining multi-robot trajectory datasets and training a single model to jointly generate trajectories for all robots. To test this approach, we evaluated *MPD-Composite*, a state-of-the-art motion planning diffusion model (MPD) (Carvalho et al., 2023) that we trained on multi-robot data we collected in two maps: Empty and Highways. We created three models: for 3, 6, and 9 robots. Each model was also given an additional guidance term that penalized collisions between robots. Across 300 tests, 50 for each map and group size, we compared MPD-Composite to MMD-xECBS (referred to as MMD). See Fig. 2 for results. The composite model achieved perfect success rates and high data adherence scores with 3 robots but struggled as the number of robots increased to 6. No valid trajectories were generated in any test with 9 agents using MPD-Composite in either map. In contrast, MMD solved all 300 problems successfully and further scaled to 40 agents in additional random tests, also producing trajectories with high data adherence scores and no collisions.

## 4.2 MMD OUTPERFORMS MAPF WITH LEARNED COST MAPS

From a search-based planning viewpoint, a compelling way to integrate motion data into multi-robot planning is by reducing MRMP to MAPF and forming cost maps to direct algorithms like ECBS (Cohen et al., 2016). This sacrifices dynamic feasibility, limiting solutions to a fixed graph, but can still offer a desirable homotopy class and be assessed for data compliance. Our second experiment set evaluates this method's potential to produce data-compliant motions (see Fig. 3). We created 180 motion planning problems, 10 per group size across three maps, and assessed two search-based methods: *A*Data-ECBS* and *A*-ECBS*. The former plans single-robot paths using A* (Nilsson, 1982) with statistical cost maps, where edge costs are lower if map dataset trajectories frequently visit those areas. The latter uses uniform costs (i.e., has no knowledge of the data) and is reported only to provide context for A*Data-ECBS's performance.

Integrating motion data into statistical cost maps shows mixed results: it improves Highways map performance by $35\%$ over A*-ECBS but reduces success rates. In other maps, it finds collision-free paths but struggles with complex motion distributions. MMD methods, by contrast, effectively generate trajectories with high data adherence and often high success rates. As shown in A.2.1 and Table 2, planning time and solution quality correlate: when A*Data-ECBS matches MMD's data adherence, their planning times align, but faster A*Data-ECBS solutions come at the cost of lower quality. We believe there is significant potential to improve the computational efficiency of MMD through parallelization. However, we have left such optimizations for future work.

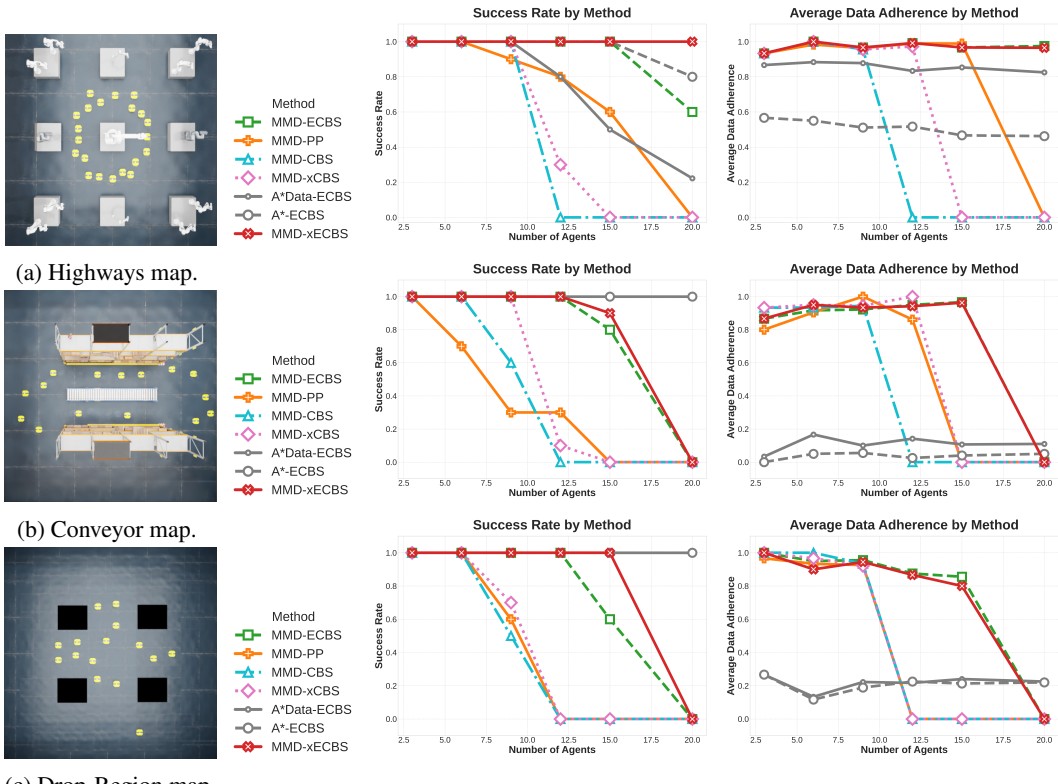

(a) Highways map.

(b) Conveyor map.

(c) Drop-Region map.

Figure 3: Analysis of success rates and data adherence scores, in randomly generated planning queries, of all MMD instantiations and a MAPF method with and without a learned cost map. The left column shows our test maps, the middle column compares success rates across 10 trials per robot count, and the right column presents the average data adherence scores.

### 4.2.1 ANALYZING MMD CONSTRAINING STRATEGIES.

Further analysis reveals a trend familiar from the MAPF literature. MMD-CBS struggles with scalability, MMD-ECBS significantly outperforms it, and accelerated versions further improve performance. MMD-PP finds efficiency in requiring only one inference pass per robot, however, because the constraints on diffusion models are soft, trajectory generation queries are not guaranteed to completely avoid higher-priority robots, and as such MMD-PP may fail to produce collision-free solutions. This is reflected in lower success rates in congested maps. In contrast, CBS-based MMD methods only failed by exceeding our 60-second runtime limit. MMD-xECBS outperformd other MMD algorithms in success rates and matched MMD-ECBS in data adherence.

To test MMD's MRMP solving ability without regard to data adherence, we created two free-space experiments focusing on robot interactions (Fig. 4). In the *circle* setup, robots move between opposite points on a circle, likely colliding at the center. In the *weave* setup (inspired by Tajbakhsh et al. (2024)), robots begin on opposite points of a square, aiming to switch places. CBS-based methods were challenged in *circle* since incrementally constraining the center region is time-consuming. MMD-PP's stronger constraints navigated around congestion and solved more problems, however, occasionally failed to produce valid solutions. In *weave*, where navigating around congestion is more difficult, CBS-based methods generated collision-free trajectories more effectively.

### 4.3 SEQUENCED DIFFUSION MODELS FOR LONG-HORIZON PLANNING

We present a feasibility study on expanding MMD algorithms to larger maps using the sequencing method described in Sec. 3.3. This technique assembles smaller local maps, each associated with a

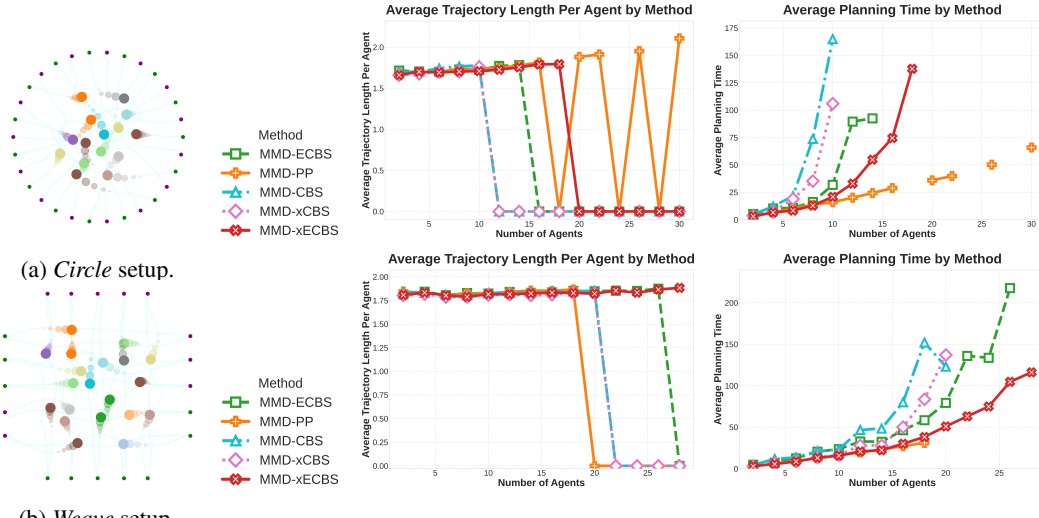

(a) *Circle* setup.

(b) *Weave* setup.

Figure 4: Scalability tests in high-congestion free-space MRMP. *Circle* (top row) asks robots to swap positions between opposite points on the perimeter. *Weave* (below), asks robots to exchange positions along uniformly spaced boundary points. Length is zero for failed problems (MMD-PP failures were due to yielding invalid solutions, and other methods failed by exceeding 240 seconds).

diffusion model, to collectively generate long-horizon, data-driven trajectories. In our experiments, we tested three MMD methods—MMD-PP, MMD-ECBS, and MMD-xECBS—in 120 trajectory generation tests, allowing a relaxed 240-second time frame. A standout feature of the sequencing method is its ability to create trajectories that follow a *task-skeleton*: passing through a series of local maps within the larger global map. For this purpose, our experiments assign a random sequence of three tasks per robot (a sequence of local maps), and randomly selects start and goal states within the first and last local maps in the sequence. The results (Fig. 5) show MMD-xECBS scaling to long planning horizons without compromising data adherence. This demonstrates MMD's capability to efficiently produce multi-robot trajectories in large environments by utilizing diffusion models trained with easily gathered data from small, local maps.

## 5 RELATED WORK

**Multi-Robot Motion Planning.** Many MRMP algorithms (Sanchez & Latombe, 2002; Solovey et al., 2016) approach the problem by treating it as a coupled system – applying sampling-based planners such as PRM (Kavraki et al., 1996) and RRT (Karaman & Frazzoli, 2011) to the composite configuration space of all robots. While this approach guarantees probabilistic completeness, it struggles to scale with the number of robots due to the exponential growth of the configuration space. Given the PSPACE-hardness of MRMP (Hopcroft & Wilfong, 1986), many practical algorithms introduce approximations to make the planning problem more tractable. Some of the most successful MRMP methods transform the problem into a multi-agent path finding (MAPF) problem (Stern et al., 2019) through state and time discretization (Li et al., 2019; Hönig et al., 2018).

*Decoupled* algorithms like prioritized planning (Erdmann & Lozano-Perez, 1987) and Leroy et al. (1999) generate robot motions one after another, fixing each plan and regarding them as dynamic obstacles in subsequent planning iterations. This allows them to quickly find solutions for large numbers of robots, but sometimes fail to find any solution even if one exists, i.e., they are often incomplete. Seeking to balance theoretical guarantees with practical efficiency, *hybrid* algorithms like CBS (Sharon et al., 2015) and its variants (Barer et al., 2014; Li et al., 2021) decouple the MRMP into two levels – high-level, in which they resolve conflicts between robots, and low-level, in which they plan motion for each robot independently. A* and its derivatives are the de facto standard for motion planning for navigation, serving as the low-level search algorithm in many of these state-of-the-art approaches, as they are efficient, complete, and bounded sub-optimal.

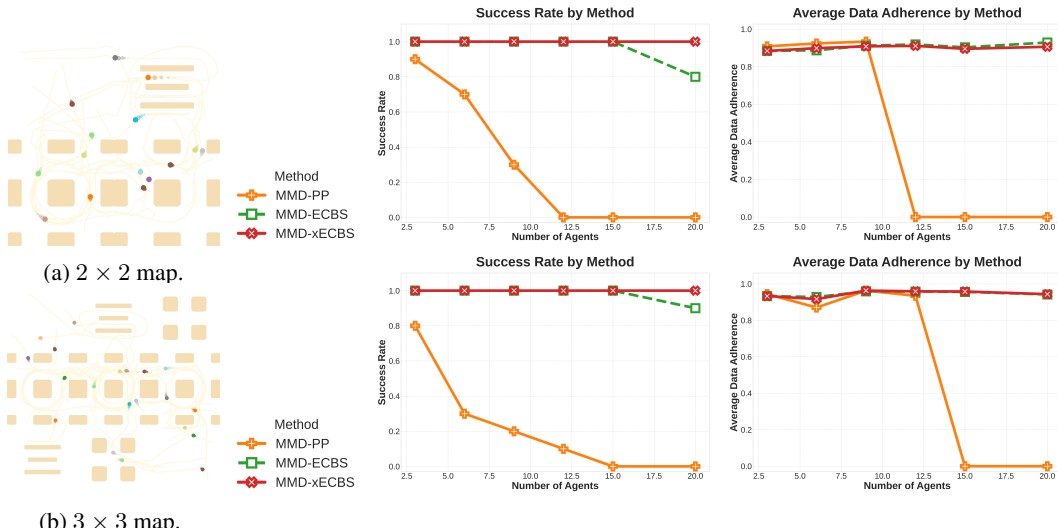

(a) $2 \times 2$ map.

(b) $3 \times 3$ map.

Figure 5: Experimental setup and results for scaling MMD to larger environments and longer planning horizons. MMD still relies on single agent data in small, local maps and does not require training of new networks for this scale-up. We evaluate three MMD variants across two large maps made of tiled local maps to cover a significantly larger area.

**Planning with Diffusion Models.** Lately, there has been a surge of interest in applying diffusion models (Sohl-Dickstein et al., 2015) to solve sequential decision making tasks, including planning and reinforcement learning (Ubukata et al., 2024). Diffuser (Janner et al., 2022) first proposed the idea of using diffusion models for trajectory planning and showed how classifier-guided sampling and image inpainting can be used for adaptation at test time. Recent works have used diffusion models as priors for single-robot motion planning (Carvalho et al., 2023), for learning visuomotor policies (Chi et al., 2024), and for offline decision-making (Ajay et al., 2023). Most of these have been limited to single-agent planning with two notable exceptions. Jiang et al. (2023) learns a joint motion distribution for multi-agent motion prediction and concurrent work by Mishra et al. (2024) uses spatial-temporal factor graphs to compose modular generative models for solving long-horizon bimanual tasks. By contrast, we focus on generating kinodynamically feasible and collision-free trajectories for dozens of robots and complex environments without learning a joint distribution.

## 6 CONCLUSION

In this paper, we present MMD, a multi-robot motion planner that learns to generate smooth collision-free trajectories for dozens of robots in complex environments. Our key contribution is showing how single robot diffusion models can be effectively combined with search algorithms to generate data-driven multi-robot trajectories. By learning only single-robot diffusion models, MMD simplifies data requirements and breaks the curse of dimensionality plaguing approaches that learn from multi-robot data. Additionally, by learning generative models of robot trajectories, MMD overcomes many of the limitations of popular model-based MAPF algorithms, such as state discretization, known cost function, and constant velocities. We believe MMD opens up exciting avenues for future work on combining the strengths of search algorithms and diffusion planning.

**Limitations.** We are excited about the potential of MMD to push forward multi-robot coordination and collaboration and offer a few avenues for future work. First is combining MMD with decentralized and windowed multi-robot planning algorithms, like Gaussian Belief Propagation (GBP) (Patwardhan et al., 2023), whose collision avoidance signals could be incorporated into single-robot diffusion guidance functions. We discuss this research direction in Sec. A.1.1. Second, we believe that the CBS-based MMD methods can be greatly accelerated, mostly through parallelization of the high-level search. Finally, we believe the frontier of MRMP lies in collaboration. Currently, MMD focuses on coordinating robots, seeking to produce collision-free data-driven trajectories. Carrying out collaborative tasks is an interesting next step.

## 7 REPRODUCIBILITY

We aspire for MMD to be easily used and extended by researchers and practitioners. To this end, we make our source code for all MMD algorithms, scripts for data generation, training, and evaluation, and evaluation maps publicly available at `https://github.com/yoraish/mmd`. This code, along with the parameter values detailed in Sec. A.4, is sufficient for reproducing the experiments and results presented in this paper. To run our code "out of the box," without dataset generation or training, we provide pre-trained models and datasets detailed in Sec. 4 with detailed instructions. For hardware and software dependencies, we specify the exact versions of libraries and tools required in our repository and also detail our hardware setup in Sec. A.4.

### ACKNOWLEDGMENTS

We thank Mohit Sharma and Ilan Mitnikov for fruitful discussions. This work was supported by the National Science Foundation under grant IIS-2328671 and grant #1955361, as well as by the Office of Naval Research (ONR) under REPRISM MURI N000142412603 and ONR grant #N00014-22-1-2592. Partial funding was also provided by a gift from Amazon and the Robotics and AI Institute.

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

# A    APPENDIX

These additional materials come to add auxiliary details on our algorithms and their implementation, alongside providing further experimental evidence that we omitted from the main text. We begin with an additional discussion on our algorithmic framework, move on to provide additional results, and conclude with implementation and baseline details.

## A.1    ADDITIONAL ALGORITHMIC DISCUSSION

In this section, we expand on a few key ways in which MMD-PP and MMD-CBS improve on their classical counterparts. These were lightly touched on in the main text. We outline them in more detail here.

**CT Node Ordering in CBS.** Traditionally, MAPF algorithms aim to find the shortest paths from starts to goals. Therefore, in CBS, the high-level nodes $N$ popped from the CT are the ones of least cost. In our case, where the data-driven objective function scoring trajectories is generally not available in practice, we instead aim to quickly find solutions that are collision-free, delegating the task of finding high-quality solutions to the diffusion models. To this end, the MMD CBS strategies choose CT nodes with the least number of conflicts to explore first, as they are more likely to lead to collision-free solutions quickly. This is similar to the algorithm GCBS-H outlined in Barer et al. (2014). Resembling their outcomes, we also observed a significant runtime improvement between prioritizing CT nodes based on their geometric quality and their collision count.

**Batch Trajectory Generation.** Planning with diffusion models has the added benefit of being able to generate a diverse set of solutions with a single inference pass (Carvalho et al., 2023). MMD makes use of this property. In contrast to the original CBS, where CT nodes $N$ store a single path for each robot within $N.\Pi$, MMD stores *trajectory batches* for each robot $\mathcal{R}_i$ within $N.\boldsymbol{\tau}$. That is, $N.\boldsymbol{\tau}^i$ may be a set of $B \in \mathbb{Z}^{>0}$ trajectories, with $B$ being a batch size. MMD-PP follows similarly, storing a batch of trajectories for each robot. When planning for $\mathcal{R}_i$, MMD generates a batch of trajectories under the currently active constraint set $N.C^i$ (see Fig. 1). Once the batch is generated, MMD iterates over the new resulting trajectories $N.\boldsymbol{\tau}^i$ and marks the one with the least collisions as the *representative trajectory* for $\mathcal{R}_i$. In MMD-PP this trajectory is used to define the placement of soft-constraints in following iterations and will be part of the solution. In MMD-ECBS, weak soft-constraints will similarly be added. All MMD CBS-based algorithms use representative trajectories to compute the number of conflicts within CT nodes. When a conflict-free CT node is found (e.g., there are no collisions between representative trajectories), MMD returns the representative trajectories of $N.\boldsymbol{\tau}$ as the solution.

### A.1.1    BEYOND FULL-HORIZON PLANNING

In this work, we focus on scenarios where it is possible to carry out centralized planning. That is, a single algorithm generates full-horizon trajectories for all robots, and robots later execute these trajectories. This formulation is common in robotics and the planning community more broadly, however, it has some limitations. For example, this setup requires that the motion of all moving obstacles be known a priori, that all robots be able to carry out their trajectories as prescribed, and that time for offline planning be available. In the real world, such information or resources may not always be available (e.g., when robots operate next to humans). In this section, we provide a brief introduction to another class of multi-robot planning algorithms—decentralized and windowed algorithms—that address some of these challenges. We also explore an exciting avenue for future work: combining these algorithms with MMD to mitigate the limitations of each approach.

The structure of decentralized and windowed multi-robot planning algorithms differs fundamentally from full-horizon planning algorithms in that they break the "plan-then-act" paradigm (Patwardhan et al., 2023; Van Den Berg et al., 2011; Okumura et al., 2019). While full-horizon planners first generate a set of trajectories for all robots and then robots execute them as prescribed, windowed algorithms instead ask each robot to plan a short trajectory for itself, execute it, observe the new state of the world, and repeat. This setup gives up on global optimality in favor of allowing faster planning cycles and decentralized computation. In that, it removes the reliance on a central planner that implicitly also assumes perfect communication between robots. Instead, planned trajectories are communicated and negotiated between neighboring robots. We believe that MMD could benefit

| Empty Map | | | | | Highways Map | | | | |
|---|---|---|---|---|---|---|---|---|---|
| $n$ | **Method** | **S**↑ | **D**↑ | **T**↓ | **A**↓ | $n$ | **Method** | **S**↑ | **D**↑ | **T**↓ | **A**↓ |
| 3 | MMD | 100% | 0.999 | 3.4 | 0.002 | 3 | MMD | 100% | 0.96 | 3.3 | 0.042 |
| | **MPD-C** | 100% | **1.000** | 2.1 | 0.010 | | **MPD-C** | 100% | **0.98** | 2.1 | 0.032 |
| 6 | **MMD** | 100% | **0.995** | 7.0 | 0.002 | 6 | **MMD** | 98% | **0.97** | 6.9 | 0.045 |
| | MPD-C | 62% | 0.331 | 2.2 | 0.142 | | MPD-C | 0% | - | - | - |
| 9 | **MMD** | 100% | **0.991** | 11.1 | 0.002 | 9 | **MMD** | 96% | **0.97** | 10.7 | 0.043 |
| | MPD-C | 0% | - | - | - | | MPD-C | 0% | - | - | - |

Table 1: Comparison of methods by number of agents in the Empty environment (left) and the Highways environment (right). **S** is the success rate (%), **D** the data adherence score (see Sec. 4), **T** is the average planning time (seconds), and **A** is average acceleration (length units/$s^2$), a proxy for smoothness. Despite being computationally efficient, the composite baseline struggles to maintain high data adherence scores. Here, MMD is MMD-xECBS.

from this structure, as it enables adaptation to dynamic environments and reduces the combinatorial complexity of MAPF algorithms by which our proposed MMD algorithms are inspired. However, the best approach for adapting MMD to decentralized and windowed settings remains unclear.

One algorithm that could shed light on how to step towards decentralized and windowed data-driven trajectory generation with MMD is Gaussian Belief Propagation Planner (GBP) (Patwardhan et al., 2023). This planner is a recently proposed decentralized and windowed algorithm for multi-robot planning that frames short-horizon planning as inference. A particular similarity between GBP and MMD is the way single-robot plans influence each other to achieve collision-free multi-robot plans. In both MMD and GBP, soft constraints are imposed on single-robot time-discretized trajectories (either full-horizon or a short-horizon window) to guide their trajectory generation processes towards favorable regions in trajectory space (e.g., collision-free, respecting dynamics, etc.). Given this similarity, it is reasonable to believe that by incorporating signals from GBP's factor graph into MMD's guidance functions, MMD could be adapted to the decentralized and windowed setting.

## A.2 EXPERIMENTAL EVALUATION: ADDITIONAL RESULTS

We provide additional quantitative and qualitative results for our first two experiment sets, outlined in Table 1, Table 2, and Fig. 6. Discussions of these experimental results are included.

### A.2.1 ADDITIONAL QUANTITATIVE RESUTLS

Our manuscript mainly evaluated algorithms based on their performance in terms of success rate and data adherence (Fig. 2, Fig. 3, and Fig. 5). While these metrics are sufficient to capture MMD's ability to consistently produce trajectories that follow an underlying motion data distribution, one may also be interested in solutions' **smoothness** and the **wall-clock time** it takes to generate those. To this end, we include results for the average acceleration per robot (column **A** in Table 1 and Table 2), as a proxy for smoothness, and the runtime for trajectory generation (column **T**).

**Average Acceleration Per Robot.** Despite showing little information, since the composite baseline quickly failed to produce valid trajectories, Table 1 offers valuable insights nonetheless. The **A** column provides a glance into the reason for composite-model's failure. Consistent with our observations, we notice that the average acceleration per robot in the 6-robot case is drastically higher for the composite model than for MMD. This behavior, visually, translates to trajectories including small loops and sharp turns. See Fig. 6 for an example. We have trained MPD-Composite to convergence using our datasets (Sec. A.6) and performed additional tuning, however, its generated motions struggled to capture the smooth, dynamically feasible transitions as the number of robots grew to 9. MMD, however, produced better and larger-scale trajectories. Table 1 shows MMD keeping low average accelerations per robot. In larger teams of robots, we notice the acceleration remaining mostly constant within each map even when the number of robots grows. This shows that robots produce similar motions within each map regardless of congestion levels—a sign of consistency that we seek as the number of robots scales.

**Trajectory Generation Runtime.** While the composite models enjoy an invariance to the number of robots, as those only require a single inference pass for multi-robot trajectory generation, this is not

| n | Method | Highways Map | | | | Conveyor Map | | | | Drop-Region Map | | | |
|---|---|---|---|---|---|---|---|---|---|---|---|---|---|
| | | S↑ | D↑ | T↓ | A↓ | S↑ | D↑ | T↓ | A↓ | S↑ | D↑ | T↓ | A↓ |
| 3 | MMD-xECBS | 100% | 0.93 | 3.6 | 0.04 | 100% | 0.87 | 3.9 | 0.10 | 100% | 1.00 | 3.7 | 0.05 |
| | MMD-ECBS | 100% | 0.93 | 3.5 | 0.04 | 100% | 0.87 | 4.5 | 0.09 | 100% | 1.00 | 3.8 | 0.05 |
| | MMD-PP | 100% | 0.93 | 4.3 | 0.04 | 100% | 0.80 | 4.2 | 0.09 | 100% | 0.97 | 4.2 | 0.05 |
| | MMD-CBS | 100% | 0.93 | 4.0 | 0.04 | 100% | 0.93 | 5.7 | 0.10 | 100% | 1.00 | 6.7 | 0.06 |
| | MMD-xCBS | 100% | 0.93 | 3.6 | 0.04 | 100% | 0.93 | 4.3 | 0.11 | 100% | 1.00 | 4.5 | 0.07 |
| | A*Data-ECBS | 100% | 0.87 | 3.4 | - | 100% | 0.03 | 0.2 | - | 100% | 0.27 | 0.2 | - |
| | A*-ECBS | 100% | 0.57 | 0.1 | - | 100% | 0.00 | 0.5 | - | 100% | 0.27 | 0.2 | - |
| 6 | MMD-xECBS | 100% | 1.00 | 7.0 | 0.05 | 100% | 0.95 | 9.3 | 0.11 | 100% | 0.90 | 7.7 | 0.06 |
| | MMD-ECBS | 100% | 1.00 | 7.3 | 0.05 | 100% | 0.92 | 13.2 | 0.09 | 100% | 0.95 | 8.4 | 0.05 |
| | MMD-PP | 100% | 0.98 | 8.9 | 0.04 | 70% | 0.90 | 8.7 | 0.09 | 100% | 0.93 | 8.4 | 0.05 |
| | MMD-CBS | 100% | 1.00 | 14.2 | 0.05 | 100% | 0.93 | 18.9 | 0.10 | 100% | 1.00 | 16.7 | 0.06 |
| | MMD-xCBS | 100% | 1.00 | 10.2 | 0.05 | 100% | 0.95 | 11.0 | 0.14 | 100% | 0.97 | 12.1 | 0.08 |
| | A*Data-ECBS | 100% | 0.88 | 4.5 | - | 100% | 0.17 | 1.1 | - | 100% | 0.13 | 0.9 | - |
| | A*-ECBS | 100% | 0.55 | 0.7 | - | 100% | 0.05 | 2.4 | - | 100% | 0.12 | 1.1 | - |
| 9 | MMD-xECBS | 100% | 0.97 | 12.7 | 0.05 | 100% | 0.93 | 14.6 | 0.11 | 100% | 0.94 | 12.9 | 0.06 |
| | MMD-ECBS | 100% | 0.97 | 15.0 | 0.05 | 100% | 0.92 | 19.6 | 0.09 | 100% | 0.96 | 15.5 | 0.06 |
| | MMD-PP | 90% | 0.96 | 13.5 | 0.05 | 30% | 1.00 | 12.4 | 0.10 | 60% | 0.93 | 12.3 | 0.06 |
| | MMD-CBS | 100% | 0.96 | 43.8 | 0.05 | 60% | 0.94 | 46.6 | 0.11 | 50% | 0.93 | 47.5 | 0.08 |
| | MMD-xCBS | 100% | 0.96 | 29.8 | 0.05 | 100% | 0.94 | 32.8 | 0.20 | 70% | 0.92 | 39.1 | 0.11 |
| | A*Data-ECBS | 100% | 0.88 | 14.0 | - | 100% | 0.10 | 2.7 | - | 100% | 0.22 | 1.3 | - |
| | A*-ECBS | 100% | 0.51 | 3.4 | - | 100% | 0.06 | 3.4 | - | 100% | 0.19 | 0.8 | - |
| 12 | MMD-xECBS | 100% | 0.99 | 15.8 | 0.04 | 100% | 0.94 | 23.1 | 0.13 | 100% | 0.87 | 22.9 | 0.07 |
| | MMD-ECBS | 100% | 0.99 | 17.6 | 0.04 | 100% | 0.95 | 28.4 | 0.10 | 100% | 0.88 | 29.6 | 0.06 |
| | MMD-PP | 80% | 0.99 | 18.4 | 0.05 | 30% | 0.86 | 16.9 | 0.10 | 0% | - | - | - |
| | MMD-CBS | 0% | - | - | - | 0% | - | - | - | 0% | - | - | - |
| | MMD-xCBS | 30% | 0.97 | 54.6 | 0.05 | 10% | 1.00 | 53.5 | 0.19 | 0% | - | - | - |
| | A*Data-ECBS | 80% | 0.83 | 19.7 | - | 100% | 0.14 | 3.8 | - | 100% | 0.22 | 4.0 | - |
| | A*-ECBS | 100% | 0.52 | 6.0 | - | 100% | 0.03 | 4.0 | - | 100% | 0.23 | 4.4 | - |
| 15 | MMD-xECBS | 100% | 0.97 | 24.3 | 0.05 | 90% | 0.96 | 38.3 | 0.14 | 100% | 0.80 | 35.7 | 0.07 |
| | MMD-ECBS | 100% | 0.97 | 29.3 | 0.05 | 80% | 0.97 | 43.7 | 0.10 | 60% | 0.86 | 43.0 | 0.06 |
| | MMD-PP | 60% | 0.99 | 23.2 | 0.05 | 0% | - | - | - | 0% | - | - | - |
| | MMD-CBS | 0% | - | - | - | 0% | - | - | - | 0% | - | - | - |
| | MMD-xCBS | 0% | - | - | - | 0% | - | - | - | 0% | - | - | - |
| | A*Data-ECBS | 50% | 0.85 | 32.0 | - | 100% | 0.11 | 14.4 | - | 100% | 0.24 | 7.7 | - |
| | A*-ECBS | 100% | 0.47 | 10.5 | - | 100% | 0.04 | 13.4 | - | 100% | 0.21 | 9.3 | - |
| 20 | MMD-xECBS | 100% | 0.96 | 46.1 | 0.05 | 0% | - | - | - | 0% | - | - | - |
| | MMD-ECBS | 60% | 0.97 | 51.8 | 0.05 | 0% | - | - | - | 0% | - | - | - |
| | MMD-PP | 0% | - | - | - | 0% | - | - | - | 0% | - | - | - |
| | MMD-CBS | 0% | - | - | - | 0% | - | - | - | 0% | - | - | - |
| | MMD-xCBS | 0% | - | - | - | 0% | - | - | - | 0% | - | - | - |
| | A*Data-ECBS | 22% | 0.82 | 43.5 | - | 100% | 0.11 | 15.6 | - | 100% | 0.23 | 15.7 | - |
| | A*-ECBS | 80% | 0.46 | 21.6 | - | 100% | 0.05 | 21.0 | - | 100% | 0.22 | 16.5 | - |

Table 2: Additional results for a subset of our MMD and MAPF evaluation. Table columns are similar to Table 1. We omit acceleration information from the MAPF methods as those plan on a grid graph and assume constant velocities.

the case for MMD. For once, all MMD algorithms begin with the sequential process of generating trajectories for all robots one at a time. The time for this operation, of course, scales linearly with the number of robots. As mentioned in our conclusion, the MMD CBS-based methods are naturally parallelized. Since the runtime of these methods is tightly related to the number of CT nodes created, which could be evaluated in parallel, doing so may drastically reduce runtimes. We leave this to future work.

**MAPF Baseline Runtime.** The trends of our timing results for the MAPF baselines shown in Table 2 are inconsistent across maps. Interestingly, the planning time for A*Data-ECBS in the Highways map was significantly higher than that of A*-ECBS. This comes with the added benefit of the produced paths better adhering to data. In the other two maps, which have more challenging underlying motion data distributions, the difference between A*Data-ECBS and A*-ECBS was insignificant, though so were the data adherence scores. It is unclear to us how to effectively incorporate demonstrations from data into MAPF solvers without compromising their ability to scale.

## A.3 ADDITIONAL QUALITATIVE RESULTS

To better capture the behavior of the various trajectory generators discussed in this paper, Fig. 6 shows a series of images of generated trajectories in two problems. We keep the number of robots low for clarity. Videos are available in our supplementary materials as well.

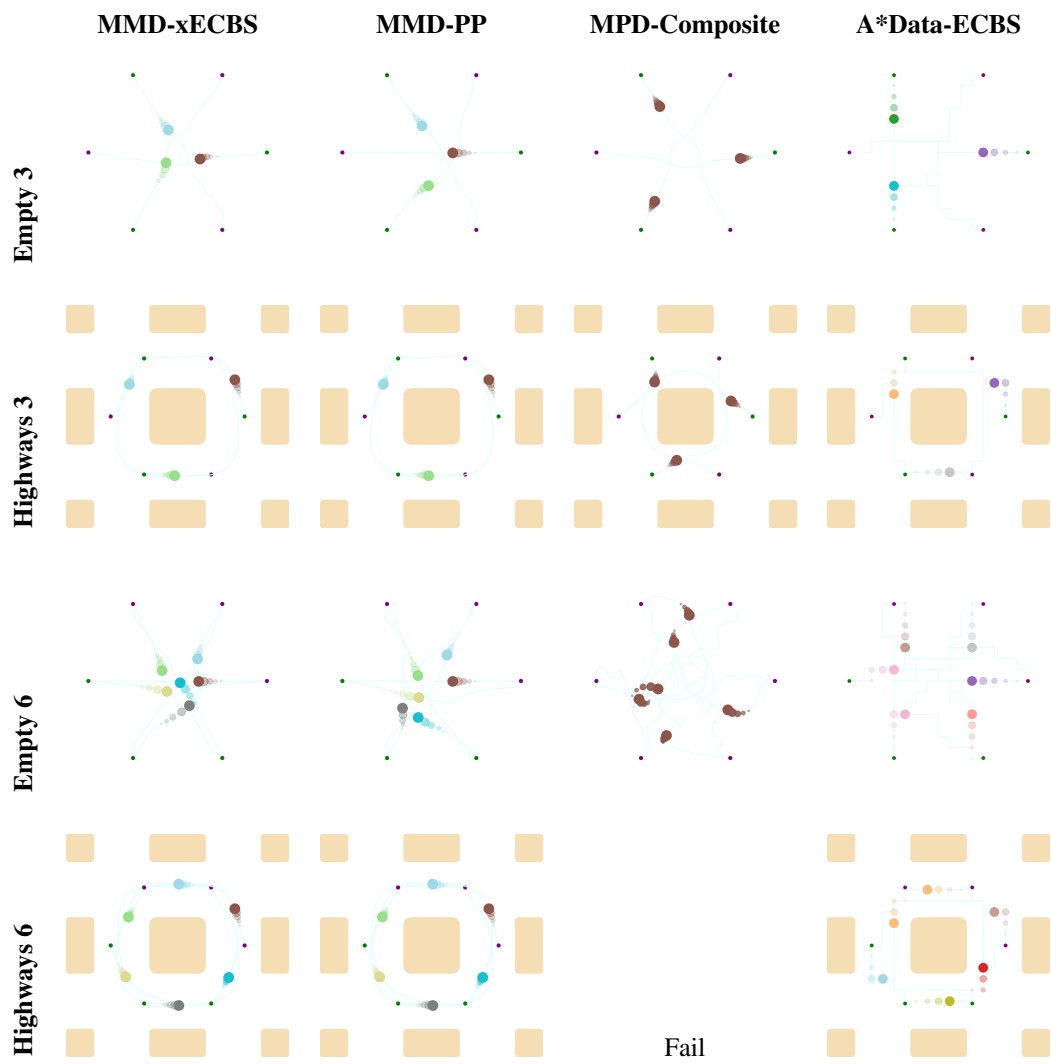

Figure 6: Visual examples of trajectories generated by MMD-xECBS, MMD-PP, MPD-Composite, and A*Data-ECBS in tests within the Empty and Highways maps. The top two rows show test cases with 3 robots, and the bottom two rows with 6. All planning problems follow the *circle* setup (Sec. 4.2) with radius 0.6 for the Highways map and 0.8 for the Empty map.

## A.4 IMPLEMENTATION DETAILS

We implemented all of our algorithms in Python and ran our experiments on a laptop with an Intel Core i9-12900H CPU, 32GB RAM (5.2GHz), and Nvidia GeForce RTX 3080Ti Laptop GPU (16 GB). We based our diffusion planning implementation on the official code of Carvalho et al. (2023) and used an exponential variance schedule. The guidance function cost components we used were $\mathcal{J}_{\text{smooth}}$ to encourage dynamically feasible trajectory generation with GPMP, $\mathcal{J}_{\text{obj}}$ for obstacle avoidance (both from Carvalho et al. (2023)), and $\mathcal{J}_{\text{c}}$ for constraint satisfaction. We set the weights $\lambda_{\text{smooth}} = 8e{-}2$, $\lambda_{\text{obj}} = 2e{-}2$, and $\lambda_{\text{c}} = 2e{-}1$ for strong soft-constraints and $\lambda_{\text{c}} = 2e{-}2$ for weak soft-constraints.

In our experiments, the size of each local map was $2 \times 2$ units, and the diameter of each disk robot was 0.1 units. The radius for CBS sphere constraints was the disk robot radius multiplied by a margin, resulting in a radius value of 0.12 units, and the time interval $\Delta t$ was 0.08 seconds (2 time steps).

### A.5 EXPERIMENTAL EVALUATION: ADDITIONAL DETAILS

This section provides details about our experimental setup, namely the data adherence scoring function $\text{cost}_{\text{data}}(\boldsymbol{\tau}^i)$ that we used to evaluate trajectories $\boldsymbol{\tau}^i$ in each map. We also include details on our baseline implementations.

#### A.5.1 DATA ADHERENCE FUNCTIONS

As discussed in Sec. 4, each of our local maps has an associated motion pattern for robots to follow when those move within it. This motion pattern is reflected in the trajectory dataset associated with each map. Sec. 4 briefly outlined the data adherence score functions for each map, and we provide more specific definitions here. We also include more details on our larger maps, including the structure of their local maps, illustrations (Fig. 7), and their adherence scores are computation method.

**Drop-Region** map (Fig. 3c) simulates package drop-off chutes. Adherence is met for $\boldsymbol{\tau}^i$, i.e., $\text{cost}_{\text{data}}(\boldsymbol{\tau}^i) = 1$, if the trajectory spends at least 25% of its duration, consecutively, in a region of radius 0.15 centered 0.15 units off the midpoint of each of the 16 chutes.

**Conveyor** map (Fig. 3b) features narrow passages with directional motion requirements. Trajectory $\boldsymbol{\tau}^i$ adheres to data if it includes a section that enters the top corridor from the right and leaves it from the left, or vice versa for the bottom corridor. There is no restriction on robots transitioning through the corridors in reverse, and no restriction on start or goal states being within the corridors. This requires robots to reason about traversal ordering.

**Highways** map (Fig. 3a), requires counter-clockwise movement around a central obstacle. The origin of the map is at the middle of the centeral obstacle. For each state transition of a trajectory $\boldsymbol{\tau}^i$, from $\boldsymbol{\tau}^i_t$ to $\boldsymbol{\tau}^i_{t+1}$, the angle between the vectors pointing from the origin to $\mathbf{q}^i_{t+1}$ and to $\mathbf{q}^i_t$ is computed. $\mathbf{q}^i_t$ is the associated configuration at $\boldsymbol{\tau}^i_t$. We define adherence to be met for $\boldsymbol{\tau}^i$ if its cumulative angle is positive, i.e., counter-clockwise.

**Empty** map (Fig. 2a) is our simplest. Trajectories $\boldsymbol{\tau}^i$, which have $H$ time steps, are scored based on the fraction of their steps that lie within a margin of a straight-line interpolation between a initial and final configurations $\mathbf{q}^i_1, \mathbf{q}^i_H$ in $\boldsymbol{\tau}^i$. Specifically, let $l$ be the distance between the first and last configurations in $\boldsymbol{\tau}^i$, and let $m$ be the number of trajectory configurations whose distance to the line $\mathbf{q}^i_H - \mathbf{q}^i_1$ is less than $\frac{l}{10}$. We define $\text{cost}_{\text{data}}(\boldsymbol{\tau}^i) := \frac{m}{H}$.

**$2 \times 2$** map (Fig. 5a) is our first instance of a *larger* map, composed by four local maps (from top left going clockwise): Empty, Conveyor, Highways, and Highways. In these larger maps, each robot is given a start and goal configuration, as well as a sequence of local maps to traverse. We can see this sequence as a form of a task-level plan. The trajectory for a robot $\mathcal{R}_i$ is generated with a single forward pass by sequencing local diffusion models, as described in Sec. 3.3. The resulting trajectory $\boldsymbol{\tau}^i$ can be seen as a concatenation of $L$ local trajectories $\boldsymbol{\tau}^{i,1}, \ldots, \boldsymbol{\tau}^{i,L}$, one for each local map. Since robots must adhere to the motion pattern prescribed by each local map they move through, we compute the total data-adherence score for a trajectory to be the average adherence across all its traversed local maps, i.e., $\text{cost}_{\text{data}}(\boldsymbol{\tau}^i) = \frac{1}{L} \sum_{l=1}^{L} \text{cost}_{\text{data}}(\boldsymbol{\tau}^{i,l})$.

**$3 \times 3$** map (Fig. 5b) is our second instance of a larger map, composed by the local maps (in row-major): Empty, Conveyor, Drop-Region, Highways, Highways, Highways, Conveyor, Drop-Region, Empty. Data adherence in all larger maps is computed similarly to the $2 \times 2$ map described above.

To mimic real-world scenarios, we have staggered the start times of robots in our large scale experiments in Sec. 4.3. There, robots began moving 10 time steps apart. Robots in motion were required to avoid static robots. Those may be stopped in high-density regions, potentially causing congestion.

### A.6 BASELINE DETAILS

Our experiments include two families of baselines: composite models and MAPF methods. Both baselines attempt to produce trajectories (or paths) that distribute according to an underlying motion pattern that is reflected in a dataset. This section comes to provide additional details regarding these baseline methods.

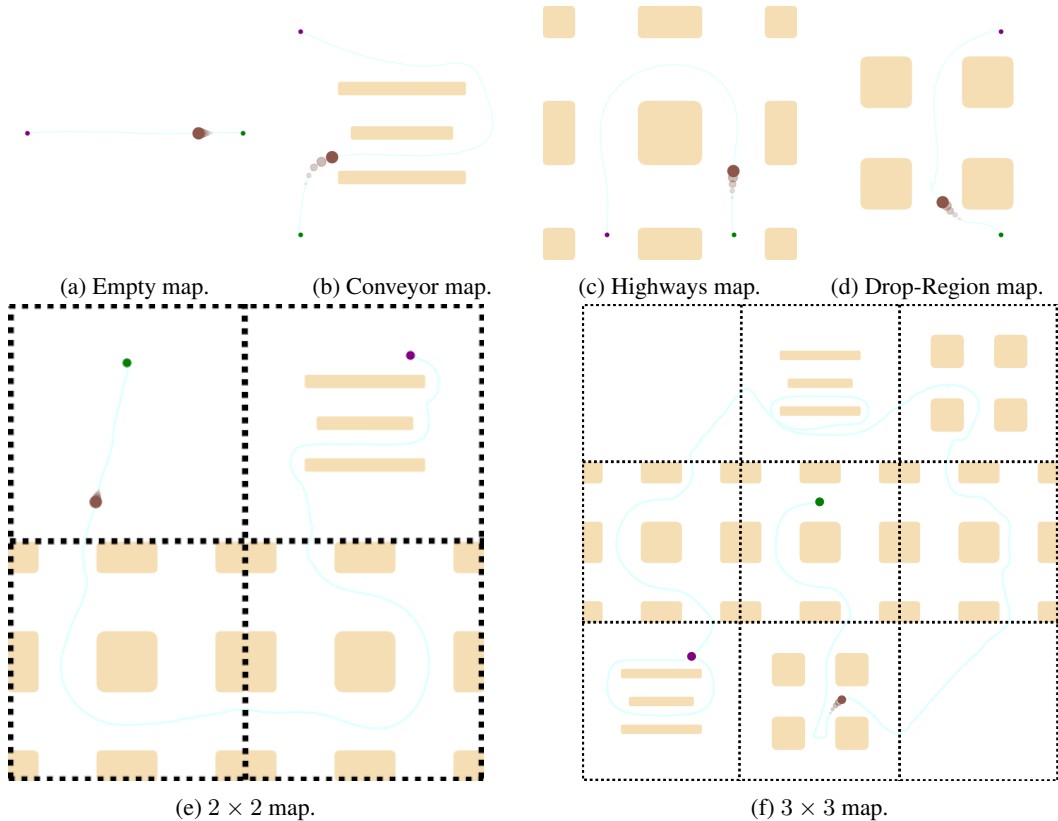

|  |  |  |  |
|---|---|---|---|
| (a) Empty map. | (b) Conveyor map. | (c) Highways map. | (d) Drop-Region map. |

(e) $2 \times 2$ map.

(f) $3 \times 3$ map.

Figure 7: Illustrations of our different maps. In the top row, we show an example trajectory that follows the data distribution prescribed by maps' underlying motion patterns. Below, we include illustrations of our $2 \times 2$ and $3 \times 3$ maps and outline the boundary between their local maps for better clarity. The larger maps include visualizations of single-robot trajectories generated by MMD with a perfect data-adherence score.

**Composite Models.** Training composite models requires two components. The first is a dataset of multi-robot motions, and the second is a model that can be trained to generate trajectories that resemble those in the data. Given that obtaining multi-robot motion datasets is generally intractable as it requires solving an MRMP problem for each datapoint, in this paper we focus on single-robot data and construct our multi-robot datasets from collision-free subsets of our single-robot datasets. It is worth noting that given that our multi-robot datasets were composed of single-robot data, the conclusions from our results is that it is difficult to learn multi-robot trajectory generation from composed single-robot data, and not that it is a uniquely challenging task in the general case. See Sec. 4.1 for a description of our composite models.

**MAPF Baselines.** Our MAPF baselines, termed A*Data-ECBS and A*-ECBS, are both similar algorithmically and only differ in the edge costs they use for single-robot graphs. We will start with describing the algorithms and then give additional details for cost map creation. The low-level planners used in these baselines are A* with a focal list mechanism, also known as $A_\epsilon^*$ (Pearl & Kim, 1982). We set the focal list bound to $1.5$ and did not allow for re-expansions in our implementation. On the high-level search, we prioritized CT nodes strictly based on their conflict count to match the strategy in MMD. All robots in our experiments travelled on a 4-connected regular grid graph with step size of $0.1$, and were allowed to move up, down, left, right, or wait on each time step, with each action incurring a unit cost in A*-ECBS. It is worth noting that the robots in our experiments are not point-robots, and so robot-robot collision checking is needed to find conflicts (e.g., two robots traversing different edges between $t$ and $t'$ may still cause an edge conflict). This affected runtime. To encode dataset demonstrations in A*Data-ECBS, we created cost maps for each evaluation map as discussed in Sec. 4.2. To do so in a given map, we iterate over all its dataset entries, and follow each trajectory, keeping track of which discretized cells (centered at states $s$) it visited. For each

cell that the trajectory visited, we find the first trajectory state outside of its center $s$, call it $s'$, and increment a count for the outgoing graph (directed) edge from $s$ that aligns best with the line between $s$ and $s'$. Eventually, we assign directed edge costs of $1 + \frac{10}{m}$ with $m$ being the number of trajectories that incremented this edge or 1 if none have. Our A*Data-ECBS and A*-ECBS implementation was done in Python.

## A.7 TRAINING AND DATASET GENERATION DETAILS

In this section, we give a brief overview of our data generation and network training processes. We note that our code for generating data, training models, and multi-robot motion planning with MMD is publicly available, and we encourage readers to consult it for exact implementation details.

**Dataset Generation:** In this work, each (local) map is associated with a dataset of trajectories. There, each data point is one, single-robot, trajectory from a random collision-free start configuration to a random collision-free goal configuration. The trajectory connecting the start and goal is discretized uniformly to 64 points such that the time between consecutive trajectory configurations is constant. Each configuration on a trajectory is complemented with velocity information too. Importantly, each dataset trajectory respects the motion pattern dictated by the map within which it is embedded. For example, trajectories in the Empty map will be straight lines, and in the Conveyor map trajectories will all pass through either of the conveyor passages. See Fig. 7 for illustrations of similar trajectories to those found in our datasets. To create the datasets, we endow each map with a motion pattern function that, given start and goal configurations, generates the critical motions that are associated with adhering to a map's underlying motion pattern. We call these motions *skill sub-paths*. For instance, in the Conveyor map, a skill sub-path will be a short sequence of configurations moving a robot throughout one of the corridors in the map. Given a skill sub-path, we create the final dataset trajectory by connecting the start configuration to the beginning of the skill sub-path, the end of the skill sub-path to the goal configurations (both with RRT-Connect (Kuffner & LaValle, 2000)), and finally smooth the resulting trajectory with a B-spline and an optimizer. This process is heavily inspired by the methodology used by Carvalho et al. (2023).

**Training procedure:** The motion planning diffusion models that we use in this work generate single-robot trajectories. Therefore, during training time, they are not required to reason about other robots. This allows us to use previously established training methodologies for motion planning diffusion models directly. In our work, we follow the same training procedure outlined in Carvalho et al. (2023), though other options can be used as well.

## A.8 RECOMMENDATIONS FOR PRACTITIONERS

We have presented five MMD variants in this paper alongside other approaches to multi-robot motion planning under learned motion distributions. Our experience with these algorithms has shed light on some of their practical strengths and weaknesses, which could be of interest to practitioners who wish to deploy or extend MMD. To this end, we offer a short set of recommendations regarding which MMD algorithms perform best in different scenarios. First, in situations with a relatively small number of robots, the differences between the MMD variants are less pronounced, as all of them solve problems relatively well. This is mostly owed to the capabilities of diffusion planning models since, in those cases, coordination is relatively easy. In such cases, we recommend using one of the CBS-based MMD algorithms since those will guarantee that a solution will be collision-free (and will normally find a solution within a short time). Among the CBS-based MMD algorithms, MMD-xECBS proved to be the best choice, as it is more efficient than the other CBS-based MMD variants and can find solutions with a similar quality.

When the number of robots increases, we can distinguish between two types of scenarios: those that are relatively *free* and those that are *cluttered*. Free scenarios are those in which the robots can easily move around each other. The reason for this could be because there are not many obstacles in the environment (e.g., our Empty map) or because the underlying learned motion patterns help avoid congestion (e.g., our Highways map). Cluttered scenarios are the opposite, where the robots are forced to move close to each other, either by the environment or by the learned motion patterns. In free scenarios, we recommend considering MMD-PP, as it is the fastest algorithm and can effectively find collision-free solutions. However, in cluttered scenarios, we recommend using MMD-xECBS

since its search over constraint-configurations showed promise in handling congestion, performing better than MMD-PP.

