# OpenReview forum: "Multi-Robot Motion Planning with Diffusion Models"
_ICLR.cc/2025/Conference — ICLR 2025 Spotlight_

### Official Review · Reviewer_WAyg · 2024-10-25

**Soundness:** 3
**Presentation:** 4
**Contribution:** 4
**Rating:** 8
**Confidence:** 4

**Summary:**

The paper addresses the challenge of collision-free multi-robot motion planning (MRMP) using diffusion models, aiming to enable the coordination of multiple robots with limited training data by leveraging single-robot trajectories. The proposed Multi-robot Multi-model planning Diffusion (MMD) method integrates single-robot diffusion models with constraint-based MAPF (multi-agent pathfinding) techniques, enabling multi-robot trajectory diffusion. MMD adapts classical MAPF strategies, including Prioritized Planning (PP) and Conflict-Based Search (CBS), to impose spatial and temporal constraints within the diffusion process, ensuring collision avoidance. An ensemble of local diffusion models is employed for long-horizon planning by sequencing trajectories across different environment contexts.

Evaluation is conducted through simulated tests across multiple scenarios and environments, including logistics-inspired settings with varying numbers of robots. Metrics used include success rates, data adherence (alignment of trajectories with data-derived patterns), and computation time. Comparisons are made with MAPF baselines and composite diffusion models trained with multi-robot data.

**Strengths:**

MAPF can be considered a hierarchical planning method for multi-robots, consisting of a high-level "negotiation" mechanism and a low-level "motion" planner.

1. It is interesting to see how the MPD [1] advantages are utilized in MAPF algorithms. Classical MAPF algorithms usually utilize RRT or PRM planners as underlying collision-avoidance planning. Indeed, utilizing a single-robot MPD model, which typically captures the local multimodality of planning behaviors as a learned smooth collision-avoidance mechanism, is a very interesting idea.

2.  The significance of MMD lies in amortization [2] low-level planning with MPD, which can infer a batch of solution efficiently (i.e., with a forward pass) without RRT or PRM subroutines. This is crucial since high-level multi-agent negotiation may induce many constraints for low-level and thus significantly speed up the overall algorithm.

3. The motivation is well-discussed, and the overall method is presented clearly.

[1] Carvalho, Joao, et al. "Motion planning diffusion: Learning and planning of robot motions with diffusion models." 2023 IEEE/RSJ International Conference on Intelligent Robots and Systems (IROS). IEEE, 2023.

[2] Amos, Brandon. "Tutorial on amortized optimization." Foundations and Trends® in Machine Learning 16.5 (2023): 592-732.

**Weaknesses:**

1. It is unfortunate that this work does not consider Multi-Robot Planning with Gaussian Belief Propagation (GBP) [3], which I believe could seamlessly integrate with the MPD framework. [3] utilizes factor-graph formulation to communicate between agents, and it is straightforward to incorporate these signals into the guidance cost term Eq. 4. In contrast, MMD-CBS formulation has combinatoric negotiation between agents inherited from classical MAPF algorithms. There should be some discussions (as future works) on integrating GBP to MMD with potential advantages (e.g., distributed planning over an arbitrary time window, removing combinatoric negotiation via message-passing over factor graph) and disadvantages (e.g., high memory cost with many factors due to increasing trajectory length and number of agents, etc.).

2. The experiment outline is thoughtful and supports the main claims with good metrics and benchmarks. However, I believe the paper would be much stronger if MMD (and its variants) were additionally compared to bare GBP [3] in these benchmarks of Sec 4.2 and the long-horizon setting of Sec 4.3.

[3] Patwardhan, Aalok, Riku Murai, and Andrew J. Davison. "Distributing collaborative multi-robot planning with Gaussian belief propagation." IEEE Robotics and Automation Letters 8.2 (2022): 552-559.

**Questions:**

1. I am curious to see how well the learning method MMD performs against bare GBP [3] as an additional baseline in highly-congested maps such as Circle and Weave setups and large maps such as 2 x 2 and 3 x 3 maps with increasing agents to 50 to further assess the scalability, with the same proposed metrics of SR and adherence. This will also shed light implicitly on how well GBP compares to classical MAPF algorithms, which benefits the MAPF community further. The code of GBP is publicly available [here](https://github.com/aalpatya/gbpplanner).

2. Please discuss the potential advantages and disadvantages of integrating GBP to MMD, as stated above.

---

> ### Author Response · Authors · 2024-11-20
> **Response to reviewer WAyg**
>
> Dear Reviewer WAyg,
>
> Thank you for your insightful and thorough comments. We are very happy to hear that you found the ideas explored in our paper to be interesting, our presentation to be clear, and our experimental analysis to be thorough. Thank you for bringing the connection between GBP and MMD to our attention. We agree that there is exciting potential for integrating these algorithms, enabling new possibilities for data-driven multi-robot planning in windowed, real-time, and decentralized settings. Such integration could enable data-driven planning in environments with a priori unknown moving obstacles and decentralized online execution–areas we find particularly exciting to explore further. However, we note that the focus of our work is on centralized long-horizon data-driven planning and hence we do not compare it with bounded horizon decentralized planners like GBP and ORCA that do not learn from data.
>
> 1. *Discussion of GBP.*
>
> We agree that discussing GBP in our paper would enrich it, and we appreciate your suggestion. Specifically, we have included a discussion in our Conclusion section highlighting the connection between GBP and MMD along with the associated advantages and challenges of combining them. Additionally, we have added a standalone appendix section (A1.1) on the prospects of using MMD in online windowed settings, including an outline of the promise that integrating it with GBP holds. Reviewer frGK has also shown interest in MMD’s ability to operate in dynamic settings–we hope that this discussion will be interesting and informative for them as well.
>
> 2. *Experimental comparison with GBP.*
>
> To facilitate a discussion of the advantages and disadvantages of integrating GBP with MMD, we have run new experiments evaluating GBP in our Circle, Weave, Conveyor, Highways, and 2x2 setups. Our experimental results certainly show promise in integrating MMD with GBP. When operating in obstacle-free maps (i.e., the Empty map in the Circle and Weave setups), GBP produced slightly shorter trajectories than MMD, with the added benefit of not having to plan the full horizon. On the other hand, GBP struggled to produce trajectories in our Conveyor and Highways maps. There, robots that attempted to move from one side of the map to the other, a motion that requires moving around a large obstacle, got stuck, were repelled off the map, or collided with obstacles (depending on the choice of hyperparameters). MMD’s access to global information (in the form of training data) could help GBP-like algorithms intelligently resolve such situations by breaking the symmetry with data-driven predictions. In our 2x2 map, where trajectories are longer, GBP performed superbly–finding collision-free trajectories for up to 40 robots.
>
> This suggests that MMD could benefit from using GBP as its coordination mechanism as it will remove the combinatorial planning phase that MMD inherits from MAPF algorithms. And GBP could benefit from having diffusion models as single-robot windowed planners (potentially imposing collision avoidance soft-constraints in the same way MMD does) since those could be better globally informed and steer windowed, data-driven, movements towards their goal even in the presence of significant clutter.
> Given the potential that lies within incorporating GBP with MMD, and the depth of analysis that such an algorithm could benefit from, we believe that reporting metrics on a comparison between MMD and GBP in our paper would do this direction injustice as it would only provide a superficial comparison between the trajectory length and success rates of learning-based, centralized, methods and a windowed, decentralized, algorithm. Therefore, in our paper we prefer to include a discussion similar to the one written above, and leave a deeper quantitative comparison between GBP, MAPF algorithms, GBP with data-driven single-robot planners, and MMD to future work.
>
> Best regards,
>
> The Authors

---

> > ### Comment · Reviewer_WAyg · 2024-11-24
> >
> > I thank the authors for their effort and insightful comments on my concerns. The comparison and future potential integrations of MMD and GBP are well-discussed. I also believe the current paper introduced a thorough idea of applying diffusion trajectory generations as low-level planners for MAPF, which is novel and significant. Hence, I will increase my score.

---

> > > ### Author Response · Authors · 2024-11-26
> > >
> > > Dear Reviewer WAyg,
> > >
> > > We would like to sincerely thank you for your support and insightful questions. We value your suggestion to consider GBP as an avenue for future work and have enjoyed the interesting analysis that it sparked. We are thrilled to see that our changes and responses have resonated with you.
> > >
> > > All the best,
> > >
> > > The Authors

---

### Official Review · Reviewer_frGK · 2024-10-31

**Soundness:** 3
**Presentation:** 4
**Contribution:** 4
**Rating:** 8
**Confidence:** 3

**Summary:**

This paper presents a novel approach for multi-robot motion planning (MRMP) that combines diffusion models with multi-agent pathfinding (MAPF) constraint strategies. This approach leverages single-robot diffusion models to plan multi-robot trajectories while avoiding collisions and scaling well to larger, complex environments. Extensive simulations demonstrate this approach’s success in generating smooth, collision-free paths with high adherence to data distribution.

**Strengths:**

This paper introduces a novel approach for multi-robot motion planning, particularly in combination MAPF with diffusion models. This approach sidesteps the challenge of multi-agent data collection by training on single-robot data, allowing the method to efficiently address the curse of dimensionality typically seen in large-scale multi-agent systems. Also, it incorporates MAPF-based constraint methods, such as Conflict-Based Search, to dynamically enforce collision-free paths.

By using an ensemble of diffusion models for each robot, the capability of scaling the approach to large environments with numerous robots, without retraining, is a notable advantage. This makes the method adaptable to a range of real-world applications.

This paper provides a solid comparison of multiple MMD variants, which helps to assess the strengths and weaknesses of different constraint strategies. This thorough analysis adds value to the paper’s contributions.

The simulations conducted in various scenarios help validate the approach. Also, the success rates and data adherence scores are well presented.

**Weaknesses:**

A major limitation is the reliance on single-robot data for policy learning. This assumption may not be sufficient for complex, dynamic environments where multi-robot interactions are essential. While multiple experiments demonstrate the effectiveness of this approach, exploring the differences between multi-agent and single-agent data, and how single-agent data can benefit cooperative multi-robot motion planning, would add valuable insight.

**Questions:**

1. This paper could benefit from a more detailed discussion of the computational overhead, especially regarding the MAPF constraint strategies and the diffusion model’s denoising process.

2. Could you further clarify the computational trade-offs between the five MMD variants? It would help to know if certain strategies are more suitable for specific environments or robot counts.

3. How effectively does the approach generalize beyond warehouse-like environments? Could it perform well in less structured or highly dynamic settings?

---

> ### Author Response · Authors · 2024-11-20
> **Response to reviewer frGK**
>
> Dear Reviewer  frGK,
>
> We sincerely thank you for your thoughtful and constructive feedback. We are encouraged by your recognition of the novelty and strengths of our paper, including the ability to scale to large environments with numerous robots without retraining, thorough comparison of different MAPF-based constraining strategies, and strong empirical performance of our method. We address your questions below.
>
> 1. *Cooperative multi-robot motion planning.*
>
> We have shown in our paper that single-robot diffusion models, when combined with MAPF-inspired constraining strategies, are sufficient to plan collision-free multi-robot trajectories in complex environments with dozens of robots. We agree that this reliance on only single-robot data may not be sufficient for cooperative multi-robot motion planning, e.g., two robots moving an object collaboratively. However, this is a much harder planning problem that is beyond the scope of this paper. Nevertheless, we believe this is an exciting direction for future research. We are interested in understanding how our method can be extended to collaborative motion planning with the minimum amount of multi-robot interaction data since it is usually much harder to collect than single-robot data. We discuss this point in the conclusion section of our paper.
>
> 2. *This paper could benefit from a more detailed discussion of the computational overhead, especially regarding the MAPF constraint strategies and the diffusion model’s denoising process.*
>
> We have added a brief discussion about the computational complexity in Section 4.2 with a reference to our detailed timing results in the appendix (A.2.1). The relationship between planning time and solution quality is intrinsically linked in our context. Our experimental results show that when A\*Data-ECBS achieves similar data adherence to MMD, the planning times are comparable. However, when A\*Data-ECBS produces solutions with lower solution quality (i.e., lower data adherence), it tends to have faster planning times.
>
> It's worth noting that there is significant potential for improving the computational efficiency of our approach through parallelization, particularly in the low-level planning phase, where each agent's path can be computed independently. However, we have left such optimizations for future work.
>
> 3. *Computational tradeoff among MMD variants.*
>
> We appreciate the suggestion to clarify trade-offs among the five MMD variants. We have added an appendix section (A.8 - Recommendations for Practitioners) that explicitly details which MMD variants are best for which scenarios (environment types and robot counts). Overall, we found MMD-xECBS to be the strongest MMD variant. It efficiently returns collision-free trajectories and scales better than other methods in cluttered maps. On the other hand, MMD-PP performs and scales better in relatively open maps with low obstacle density. However, it struggles in congested environments and does not guarantee collision-free trajectories.
>
> 4. *Generalization beyond warehouse-like environments to less structured and dynamic settings.*
>
> We picked the warehouse domain for experimental evaluation due to its significant real-world importance. However, MMD is not limited to such environments and should perform well in other scenarios too. For example, we are very interested in applying MMD to multi-robot-arm motion planning in our future work.
>
> While the focus of this paper is on motion planning in the presence of static obstacles, we believe that extending MMD to more dynamic settings (e.g., robots sharing space with humans) is possible and an exciting direction for future work. One possibility is to incorporate MMD into an MPC-style execution mechanism, wherein the robots frequently replan and execute the initial part of the plan. However, it may be too slow to plan all the way to the goal using this strategy. Hence, the robots may have to plan over a bounded time window, for example, by integrating MMD with GBP as suggested by reviewer WAyg. We have added an appendix section (A1.1) on the prospects of using MMD in online windowed settings (suitable for dynamic environments), including an outline of the promise that integrating it with GBP holds.
>
> Best regards,
>
> The Authors

---

> > ### Comment · Reviewer_frGK · 2024-11-22
> >
> > Thanks for your detailed response. I will keep my positive recommendation.

---

> > > ### Author Response · Authors · 2024-11-26
> > >
> > > Dear Reviewer frGK,
> > >
> > > We would like to sincerely thank you for your continued support and are thrilled to see that our changes and responses have resonated with you.
> > >
> > > All the best,
> > >
> > > The Authors

---

### Official Review · Reviewer_wKu8 · 2024-11-01

**Soundness:** 3
**Presentation:** 2
**Contribution:** 3
**Rating:** 6
**Confidence:** 4

**Summary:**

This paper focuses on the collision-free multi-robot planning problem. They combine single-robot diffusion planning models with multi-agent path finding (MAPF) algorithms. They leverage different MAPF algorithms to generate spatio-temporal constraint functions and use these functions to guide diffusion planning models to generate soft-constraint multi-robot trajectories.

**Strengths:**

* The questions of multi-robot motion planning that the paper aims to solve is meaningful and it has many real-world applications like automated warehouses.
* The method proposed by the paper is diretly and simple. They leverage state-of-art single-robot diffusion planning methods with classifical MAPF algorithms to solve multi-robot motion planning problems.
* They also evaluate their approaches in a variety of simulated scenarios and show better performance. They also conduct rich ablation studies including planning time, different strategies.

**Weaknesses:**

* The novelty of the method seems limited. The paper seems simply combine diffusion planning methods and classifical MAPF algorithms. In the method section (3.2), It's hard to judge which parts are from classifical MAPF algorithms, which parts are contributed by the paper to adapt MAPF algorithms into diffusion planning methods.
* Lack of some experimental details and results. See questions.
* The methods are not very efficient when the number of agents increases.

**Questions:**

* How is the single-robot diffusion models training? Which dataset? Is it a single-robot dataset or a multi-robot dataset?
* Search-based algorithms (A*) achieve a higher success rate compared to MMD. Is it fair to compare search-based algorithms at data adherence scores since search-based algorithms are not learned from dataset?
* The performances of MPD are very low even if the number of agents is less than 5. Is it normal?

---

> ### Author Response · Authors · 2024-11-20
> **Response to reviewer wKu8 (part 1/2)**
>
> Dear Reviewer wKu8,
>
> Thank you for your valuable feedback and questions. We are glad that you appreciate the real-world importance of multi-robot motion planning, the significant improvement provided by our method in a variety of scenarios, and our thorough ablation studies. We address your questions below.
>
> 1. *The novelty of the method seems limited.*
>
> To the best of our knowledge, our algorithm is the first to scale diffusion planning to dozens of robots while only using readily available single-robot data. We make two central novel contributions: First, we show through extensive experimental evaluation that conditional generative models (e.g., diffusion models) struggle in directly learning to generate multi-robot trajectories due to the exponential blow-up in the configuration space with an increasing number of robots. Second, we show that it is possible to circumvent this curse of dimensionality by using diffusion models as low-level planners in coordination algorithms inspired by many classical MAPF algorithms. As far as we know, this connection has not been established before and opens the door for future work on this intersection. In fact, reviewer WAyg has already suggested that another state-of-the-art algorithm, Gaussian Belief Propagation (GBP), could potentially be integrated with our framework. This underscores the applicability of our method to endow a wide range of multi-robot coordination algorithms with data-driven trajectory generation capabilities.
>
>
> 2. *In the method section (3.2), It's hard to judge which parts are from classifical MAPF algorithms, which parts are contributed by the paper to adapt MAPF algorithms into diffusion planning methods.*
>
> Thank you for drawing our attention to this. We agree that Section 3.2 could be clearer in distinguishing between the components in MMD that are derived from established MAPF algorithms and our own contributions in adapting these algorithms to diffusion planning. To address this, we have clarified Section 3.2 to include that the only aspect directly inherited from MAPF algorithms is the logic for determining the placement of strong constraints, whether through search or prioritization. In more detail, other elements such as imposing MAPF constraints on diffusion models via guidance functions, translating the focal-list mechanism to sets of weak and strong constraints (as seen in MMD-ECBS and MMD-xECBS), and reusing previous computational efforts in CBS via short noising-denoising processes (in MMD-xCBS and MMD-xECBS), are novel contributions to multi-robot planning. We additionally demonstrate in our experimental analysis and Table 2 (Appendix) that applying Conflict-Based Search (CBS)—arguably the most widely used MAPF algorithm—to coordinate diffusion-based planners often fails to solve many planning problems within practical runtimes. Therefore, our work dives deeper and contributes more advanced coordination algorithms for diffusion models, going beyond a combination of MAPF and diffusion planning techniques.
>
> 3. *The methods are not very efficient when the number of agents increases.*
>
> Scaling algorithms to large numbers of robots is a central challenge in multi-robot motion planning, mostly due to the exponential growth of the size of the search space with increasing numbers of robots. Compared with MPD, which is unable to plan for more than 6 robots, our approach can effectively plan for dozens of robots and establishes a new state-of-the-art. Our paper is only a first step towards scalable and efficient multi-robot diffusion planning and hence focuses on presenting and thoroughly analyzing our novel algorithmic framework. MMD is by no means bounded by its current runtimes and improvements in diffusion planning will directly translate to improvements in MMD. We agree with you that there is also significant room for further algorithmic optimization and we leave that for future work.
>
> 4. *How is the single-robot diffusion models training? Which dataset? Is it a single-robot dataset or a multi-robot dataset?*
>
> We agree that adding details on our training and dataset creation processes would help make our work more accessible. To this end, we outline these here and added a section to our paper (A.7) with appropriate details. Our code will also be shared along with the final paper and include our training, dataset creation, and inference scripts together with other relevant code.
>
> **Dataset creation:** All of our datasets are single-robot datasets showcasing trajectories that follow map-specific motion patterns. For example, the Empty map has straight-line trajectories, and the Conveyor map has trajectories that pass through either of the conveyor passages. Each data point is a single trajectory. We create our datasets by connecting random start and goal configurations to map-specific sub-paths with RRT-Connect and then smooth them with an optimizer.
>
> (continued)

---

> > ### Author Response · Authors · 2024-11-20
> > **Response to reviewer wKu8 (part 2/2)**
> >
> > **Training procedure:** We follow a similar training scheme to the one proposed in MPD (Carvalho et al. 2023). Specifically, we train a temporal U-Net to predict the noise applied to trajectory samples after $k$ noising steps. We use an exponential variance schedule for noise application. For more details, we recommend consulting our code (we omit it for now to maintain anonymity). In the meantime, please refer to the code from MPD: https://github.com/jacarvalho/mpd-public/blob/main/scripts/train_diffusion/train.py.
> >
> > 5. *Search-based algorithms (A\*) achieve a higher success rate compared to MMD. Is it fair to compare search-based algorithms at data adherence scores since search-based algorithms are not learned from dataset?*
> >
> > In our analysis, we include two variants of A\* search: A\*-ECBS and A\*Data-ECBS. The first does not have access to datasets and the second does, in the form of a learned costmap. As you pointed out, it would be unfair to directly compare A\*-ECBS with MMD in terms of data adherence. We only include it to contextualize the performance of A\*Data-ECBS, analyzing how incorporating data into search-based MRMP algorithms affects their performance relative to an uninformed baseline. We compare the informed version, A\*Data-ECBS to MMD to further understand which learning method is more suitable for data-driven multi-robot trajectory generation. Our results emphasize that using diffusion models to inform trajectory generation is superior to learning cost maps in A\* search.
> >
> > 6. *The performances of MPD are very low even if the number of agents is less than 5. Is it normal?*
> >
> > The poor performance of MPD with more than 5 agents is not surprising since MPD tries to learn multi-agent trajectories directly in the high dimensional composite state space of all the agents. This state space grows exponentially with the number of agents and so does the sample complexity of MPD. Similar observations have also been made in the multi-agent motion planning literature where algorithms that plan directly in the composite state space of all the agents scale poorly [1].
> >
> > Best regards,
> >
> > The Authors
> >
> > 1. Atias, Aviel, et al. "Effective metrics for multi-robot motion-planning." The International Journal of Robotics Research 37.13-14 (2018): 1741-1759.

---

> ### Comment · Area_Chair_Rcdm · 2024-11-25
>
> Dear Reviewer,
>
> Please provide feedback to the authors before the end of the discussion period, and in case of additional concerns, give them a chance to respond.
>
> Timeline: As a reminder, the review timeline is as follows:
>
> November 26: Last day for reviewers to ask questions to authors.
>
> November 27: Last day for authors to respond to reviewers.

---

> ### Author Response · Authors · 2024-11-26
>
> Dear Reviewer wKu8,
>
> We are available to answer any questions you may have and look forward to continuing the discussion with you!
>
> All the best,
>
> The Authors

---

### Official Review · Reviewer_AAKU · 2024-11-02

**Soundness:** 4
**Presentation:** 4
**Contribution:** 3
**Rating:** 8
**Confidence:** 4

**Summary:**

This paper presents a multi-robot motion planning algorithm leveraging diffusion models and MAPF algorithms. In the proposed MMD algorithm, a conditional diffusion model is used to generate collision-free trajectories for the robot, and MAPF provides the logic to impose constraints on robots such that it scales in a multi-agent setting. The paper benchmarks with A* search and tests it on multiple multiagent tasks with varying complexity. The results show good success rate and data adherence for the proposed methods.

**Strengths:**

1. This paper provides a very thorough discussion of how to combine MAPF algorithms with diffusion models and demonstrate its ability to scale to multi-agent settings.
2. The experiment setting is diverse with varying levels of difficulties.
3. The writing is clear and structured and the paper is easy to read.

**Weaknesses:**

1. It would be good to discuss other search algorithms in multi-agent motion planning or at least discuss why only A* is chosen
2. In Sec. 4.2, it would be good to discuss the computational complexity between A* and MMD.
3. The explanation of the motion pattern is insufficient. Then it becomes a bit confusing in Sec 3.3 how this task-relevant local diffusion model works and also in Sec. 4 what the data adherence means.

**Questions:**

1. Can you explain more about what the motion pattern is? Also how this task-relevent local diffusion works and how data adherence works based on this.
2. In Figure 3, do you have an intuition as that why A* methods maintain a good success rate for (b) and (c) even though the agent number keeps increasing? Is there any tradeoff with the computational efficiency compared to MMD?
3. Is there any other search algorithm other than A* that should we consider? Can have a closer connection between the related work and the baseline methods.

**Details Of Ethics Concerns:**

None.

---

> ### Author Response · Authors · 2024-11-20
> **Response to reviewer AAKU (part 1/2)**
>
> Dear Reviewer AAKU
>
> Thank you for your constructive feedback. We are glad that you found the paper easy to understand and our experiments thorough. We are grateful for your questions and comments. We have revised the paper to incorporate your feedback and respond to your questions below.
>
> 1. *Can you explain more about what the motion pattern is? Also how this task-relevent local diffusion works and how data adherence works based on this.*
>
> We agree that better defining motion patterns and local diffusion models would strengthen our paper. We clarify this both here and also in the paper.
>
> A motion pattern represents a trajectory distribution optimizing a hidden cost function defined by a specific task dataset. For example, near a conveyor belt, we can define a motion pattern requiring robots to pass through either the top corridor right-to-left or the bottom corridor left-to-right. Regarding the local diffusion models,  our local diffusion approach involves decomposing a large environment into smaller regions. The main idea is to learn local diffusion models for each region and compose them during planning. This makes learning more sample efficient and leads to better generalization via composition.  We have added those explanations in section 3.3 and have provided more detailed explanations in Appendix A.5.1, including expanded details on data adherence score computation and extended examples in our Conveyor, Empty, Highways, and Drop-Region maps, as well as our 2x2 and 3x3 configurations. Readers could find visualizations there demonstrating these motion patterns and their adherence across different map configurations.
>
> 2. *In Figure 3, do you have an intuition as that why A\* methods maintain a good success rate for (b) and (c) even though the agent number keeps increasing? Is there any tradeoff with the computational efficiency compared to MMD?*
>
> Thank you for that question –  it probes core differences between our proposed MMD method and the A\*-ECBS and A\*Data-ECBS baselines. MMD aims to generate trajectories that maintain high data adherence, while A\*-ECBS and A*Data-ECBS show a different planning strategy that impacts data adherence. Namely, A\*-ECBS does not take into account the underlying cost of following motion patterns and, as such, opts for just finding a collision-free solution. Thus, its success rate in finding a solution is higher since it easily avoids congested regions that are important for data-adherence. In high-congestion regions like the conveyor (b) and drop-region (c), we also observe a different behavior between MMD and A\*Data-ECBS. While MMD maintains precise motion patterns (quantified by high data-adherence scores), A\*Data-ECBS redistributes robots to less congested areas– allowing it to maintain a relatively consistent success rate as agent numbers increase at the cost of poor data-adherence scores. The key difference between MMD and A\*Data-ECBS baseline is that MMD prioritizes data adherence, whereas A\*Data-ECBS balances trajectory length, fast conflict resolution, and data adherence in a way that our experimental results suggested was effective for finding valid solutions but less so for finding data-adhering ones.
>
> 3. *In Sec. 4.2, it would be good to discuss the computational complexity between A\* and MMD.*
>
> We have added a brief discussion about the computational complexity in Section 4.2 with a reference to our detailed timing results in the appendix (A.2.1). The relationship between planning time and solution quality is intrinsically linked in our context. Our experimental results show that when A\*Data-ECBS achieves similar data adherence to MMD, the planning times are comparable. However, when A\*Data-ECBS produces solutions with lower solution quality (i.e., lower data adherence), it tends to have faster planning times.
>
> It's worth noting that there is significant potential for improving the computational efficiency of our approach through parallelization, particularly in the low-level planning phase, where each agent's path can be computed independently. However, we have left such optimizations for future work.

---

> > ### Author Response · Authors · 2024-11-20
> > **Response to reviewer AAKU (part 2/2)**
> >
> > 4. *Is there any other search algorithm other than A\* that should we consider? Can have a closer connection between the related work and the baseline methods.*
> >
> > In this work, we focus on CBS and prioritization algorithms for decoupling the multi-agent motion planning problem into two levels of planning – high-level, in which we resolve conflicts between robots, and low-level, in which we plan motion for each robot independently. A\* is the de facto standard for motion planning for navigation, serving as the low-level search algorithm in many state-of-the-art approaches, including CBS, ECBS, EECBS, xECBS, and PBS [1-5], as it is efficient (particularly in low-dimensional state spaces), complete, and bounded sub-optimal.
> >
> > While alternative approaches like PRM and RRT exist and are valuable in certain scenarios (especially in high-dimensional spaces), they are less suitable for our specific context. Grid maps provide an efficient representation of our problem space. In some cases, particularly in higher-dimensional spaces, PRM is used to create a roadmap, and A\* is then used to search over it.
> >
> > We have revised our related work (section 5 - Multi-Robot Motion Planning) to clarify those points and that A\* is the common choice for MAPF algorithms in a CBS-style two-level planning framework. As you suggested, we believe that this strengthens the connection between the related work and baseline methods by explicitly discussing how our approach builds upon the established practice of using A\* in MAPF. Thank you again for raising this point.
> >
> > Best regards,
> >
> > The Authors
> >
> > 1. Guni Sharon, Roni Stern, Ariel Felner, and Nathan R Sturtevant. Conflict-based search for optimal multi-agent pathfinding. Artificial Intelligence, pp. 40–66, 2015.
> > 2. Max Barer, Guni Sharon, Roni Stern, and Ariel Felner. Suboptimal variants of the conflict-based search algorithm for the multi-agent pathfinding problem. In International Symposium on Combinatorial Search, pp. 19–27, 2014.
> > 3. Jiaoyang Li, Wheeler Ruml, and Sven Koenig. Eecbs: A bounded-suboptimal search for multi-agent path finding. In AAAI Conference on Artificial Intelligence, pp. 12353–12362, 2021.
> > 4. Yorai Shaoul, Itamar Mishani, Maxim Likhachev, and Jiaoyang Li. Accelerating search-based planning for multi-robot manipulation by leveraging online-generated experiences. In International Conference on Automated Planning and Scheduling, 2024a.
> > 5. Hang Ma, Daniel Harabor, Peter J. Stuckey, Jiaoyang Li, and Sven Koenig. "Searching with consistent prioritization for multi-agent path finding." In Proceedings of the AAAI conference on artificial intelligence, vol. 33, no. 01, pp. 7643-7650. 2019.

---

> ### Comment · Reviewer_AAKU · 2024-11-22
>
> I appreciate your kind response to my questions, and the additional material in the paper helps further explain the content. Thus, I raised my score to 8. Great paper!

---

> ### Author Response · Authors · 2024-11-26
>
> Dear Reviewer AAKU,
>
> We would like to sincerely thank you for your support. We have made our best effort to improve our paper and address your concerns, and we are thrilled to see that our changes and responses have resonated with you.
>
> All the best,
>
> The Authors

---

### Author Response · Authors · 2024-11-20
**Summary of response for all reviewers and area chair**

We would like to thank the reviewers for their thorough and insightful comments. We are greatly encouraged by the positive feedback from all the reviewers. The reviewers found our contributions to be novel and interesting and appreciated the practical importance of the problem addressed by our paper. We are also glad the reviewers liked our presentation and extensive experimental evaluation.

We address the points and questions raised across the reviews:

1. **Distinction of contributions** (wKu8): We have revised section 3.2 and explicitly distinguished between components motivated by MAPF algorithms and our novel contribution – the only aspect directly inherited from MAPF is the logic for determining the placement of collision avoidance constraints, whether through search or prioritization. Our novel contributions include imposing MAPF constraints on diffusion models via guidance functions, translating focal-list mechanisms to constraint sets, and reusing computational efforts in CBS via short noising-denoising processes. More broadly, to the best of our knowledge, our algorithm is the first to scale diffusion planning to dozens of robots using only readily available single-robot data: a contribution we believe is novel in the field.

2. **Computational efficiency and scaling** (AAKU, wKu8, frGK): While our approach represents a significant advancement in scaling diffusion models to multi-robot planning (dozens compared to MPD's limit of 6 robots), we acknowledge there is room for improvement. To better analyze the efficiency of our proposed method we have expanded the discussion of computational complexity in Section 4.2 to cover a detailed experimental analysis found in the appendix (A.2.1). Furthermore, we added a "Recommendations for Practitioners" section (A.8) detailing which MMD variants are best suited for different scenarios.

3. **Training and dataset details** (wKu8): We have added an appendix section (A.7) detailing our training procedures and dataset creation process. All our datasets are single-robot datasets showcasing trajectories that follow map-specific motion patterns. Our code release will include training, dataset creation, and inference scripts.

4. **Integration with other methods and consideration of dynamic environments** (WAyg, frGK): Following WAyg’s suggestion, we have discussed the potential integration of MMD with Gaussian Belief Propagation (GBP) methods in our conclusion and added an appendix section on using MMD in online windowed settings. Our new experiments suggest promising complementary strengths between MMD and short-horizon planners. This discussion and results are also related to the question of frGK, regarding the potential of extending our method for dynamic environments.

**Additional Clarifications:**
- We have better defined motion patterns and local diffusion models in section 3.3 (AAKU).
- We have expanded our related work to clarify why A\* is the common choice for MAPF algorithms in our context (AAKU).
- We have added visualizations demonstrating motion patterns and their adherence across different map configurations (AAKU).

We want to thank the reviewers again for their insightful feedback and for guiding us toward a revision that strengthened our paper while maintaining its core contributions.

---

### Meta-Review · Area_Chair_Rcdm · 2024-12-22

**Metareview:**

This paper introduces the Multi-robot Multi-model planning Diffusion (MMD) method, which addresses a key challenge in multi-robot motion planning (MRMP) by leveraging single-robot diffusion models combined with Multi-Agent Path Finding (MAPF) constraints. The proposed approach scales diffusion-based motion planning to dozens of robots by imposing collision avoidance constraints derived from MAPF strategies like Conflict-Based Search (CBS). The authors demonstrate the effectiveness of this framework in generating collision-free, data-adherent trajectories across various logistics-inspired simulated environments. By efficiently utilizing single-robot data, the work establishes a new benchmark for scalable diffusion-based planning in multi-agent scenarios.

The paper’s novelty lies in its integration of diffusion models with MAPF strategies, addressing the exponential complexity inherent in multi-robot systems without requiring multi-agent datasets. The method is validated through extensive experiments across diverse environments, including different robot densities and map configurations, showcasing strong empirical performance. The authors also provide thorough ablations and comparisons with relevant baselines, supporting the robustness and scalability of their approach. The clarity and presentation of the paper, including detailed methodological explanations and visualizations, further contribute to its strengths.

The paper has some limitations. The experiments are primarily focused on structured, logistics-inspired environments, leaving the generalization to dynamic or less structured settings underexplored. While the proposed method significantly improves scalability, there is room for optimization in computational efficiency, particularly as the number of robots increases. Despite these minor weaknesses, the paper’s contributions are significant. It provides a novel solution to a critical problem in robotics, with strong empirical results and clear potential for future research.

**Additional Comments On Reviewer Discussion:**

The discussion and rebuttal period were productive, with the authors addressing several key concerns raised by reviewers. Reviewer wKu8 noted the need to distinguish novel contributions from MAPF-derived components. The authors clarified this distinction in Section 3.2, explicitly highlighting the unique aspects of their method. The computational efficiency was another point of concern for reviewers wKu8 and frGK, with the authors adding detailed analyses in Section 4.2 and the appendix A.2.1, which the reviewers found insightful.

Reviewer WAyg suggested integrating the method with Gaussian Belief Propagation (GBP). While direct experiments with GBP were not included, the authors provided a thoughtful discussion on the potential advantages and challenges of such integration in Appendix A1.1, satisfying the reviewer’s concern. Additionally, questions regarding training procedures and dataset creation were addressed with the inclusion of Appendix A.7, offering detailed explanations that reviewers appreciated.

The authors’ efforts to improve the paper were well-received, with some reviewers increasing their scores after the rebuttal. The additional clarifications, experiments, and thoughtful responses strengthened the overall quality of the paper, solidifying the consensus that it is a valuable contribution to the field.

---

### Decision · Program_Chairs · 2025-01-22

Accept (Spotlight)